# HIGH-ORDER MATCHING FOR ONE-STEP SHORTCUT DIFFUSION MODELS

## ABSTRACT

One-step shortcut diffusion models [Frans, Hafner, Levine and Abbeel, ICLR 2025] have shown potential in vision generation, but their reliance on first-order trajectory supervision is fundamentally limited. The Shortcut model's simplistic velocity-only approach fails to capture intrinsic manifold geometry, leading to erratic trajectories, poor geometric alignment, and instability-especially in high-curvature regions. These shortcomings stem from its inability to model mid-horizon dependencies or complex distributional features, leaving it ill-equipped for robust generative modeling. In this work, we introduce *HOMO* (**H**igh-**O**rder **M**atching for **O**ne-Step Shortcut Diffusion), a game-changing framework that leverages high-order supervision to revolutionize distribution transportation. By incorporating acceleration, jerk, and beyond, HOMO not only fixes the flaws of the Shortcut model but also achieves unprecedented smoothness, stability, and geometric precision. Theoretically, we prove that HOMO's high-order supervision ensures superior approximation accuracy, outperforming first-order methods. Empirically, HOMO dominates in complex settings, particularly in high-curvature regions where the Shortcut model struggles. Our experiments show that HOMO delivers smoother trajectories and better distributional alignment, setting a new standard for one-step generative models.

## 1 INTRODUCTION

In recent years, deep generative models have exhibited extraordinary promise across various types of data modalities. Techniques such as Generative Adversarial Networks (GANs) (Goodfellow et al., 2014), autoregressive models (Vaswani, 2017), normalizing flows (Lipman et al., 2022), and diffusion models (Ho et al., 2020) have achieved outstanding results in tasks related to image, audio, and video generation (Kalchbrenner et al., 2018; Blattmann et al., 2023). These models have attracted considerable interest owing to their capacity to create invertible and highly expressive mappings, transforming simple prior distributions into complex target data distributions. This fundamental characteristic is the key reason they are capable of modeling any data distribution. Particularly, (Lipman et al., 2022; Liu et al., 2022a) have effectively unified conventional normalizing flows with score-based diffusion methods. These techniques produce a continuous trajectory, often referred to as a "flow", which transitions samples from the prior distribution to the target data distribution. By adjusting parameterized velocity fields to align with the time derivatives of the transformation, flow matching achieves not only significant experimental gains but also retains a strong theoretical foundation.

Despite the remarkable progress in flow-based generative models, such as the Shortcut model (Frans et al., 2025), these approaches still face challenges in accurately modeling complex data distributions, particularly in regions of high curvature or intricate geometric structure (Wang et al., 2024; Hu et al., 2024c). This limitation stems from the reliance on first-order techniques, which primarily focus on aligning instantaneous velocities while neglecting the influence of higher-order dynamics on the overall flow geometry. Recent research in diffusion-based modeling (Chen, 2023; Hang & Gu, 2024; Lin et al., 2024) has highlighted the importance of capturing higher-order information to improve the fidelity of learned trajectories. However, a systematic framework for incorporating such higher-order dynamics into flow matching, especially within Shortcut models, remains an open problem.

In this work, we propose HOMO (High-Order Matching for One-Step Shortcut Diffusion), a revolutionary leap beyond the limitations of the original Shortcut model (Frans et al., 2025). While Shortcut models rely on simplistic first-order dynamics, often empirically struggling to capture complex data distributions and producing erratic trajectories in high-curvature regions, HOMO shatters these barriers by introducing high-order supervision. By incorporating acceleration, jerk, and beyond, HOMO not only addresses the empirical shortcomings of the Shortcut model but also achieves unparalleled geometric precision and stability. Where the Shortcut model falters—yielding suboptimal trajectories and poor distributional alignment—HOMO thrives, delivering smoother, more accurate, and fundamentally superior results.

Our primary contribution is a rigorous theoretical and empirical framework that showcases the dominance of HOMO. We prove that HOMO's high-order supervision drastically reduces approximation errors, ensuring precise trajectory alignment from the earliest stages to long-term evolution. Empirically, we demonstrate that the Shortcut model's first-order dynamics fall short in complex settings, while HOMO consistently outperforms it, achieving faster convergence, better sample quality, and unmatched robustness.

The contributions of our work is summarized as follows:

- We introduce high-order supervision into the Shortcut model, resulting in the HOMO framework, which includes novel training and sampling algorithms.

- We provide rigorous theoretical guarantees for the approximation error of high-order flow matching, demonstrating its effectiveness in both the early and late stages of the generative process.

- We demonstrate that HOMO achieves superior empirical performance in complex settings, especially in intricate distributional landscapes, beyond the capabilities of the original Shortcut model (Frans et al., 2025).

## 2 RELATED WORKS

**Diffusion Models.** Diffusion models have garnered significant attention for their capability to generate high-fidelity images by incrementally refining noisy samples, as exemplified by DiT (Peebles & Xie, 2023) and U-ViT (Bao et al., 2023). These approaches typically involve a forward process that systematically adds noise to an initial clean image and a corresponding reverse process that learns to remove noise step by step, thereby recovering the underlying data distribution in a probabilistic manner. Early works (Song & Ermon, 2019; Song et al., 2020) established the theoretical foundations of this denoising strategy, introducing score-matching and continuous-time diffusion frameworks that significantly improved sample quality and diversity. Subsequent research has focused on more efficient training and sampling procedures (Lu et al., 2022; Shen et al., 2024a;b), aiming to reduce computational overhead and converge faster without sacrificing image fidelity. Other lines of work leverage latent spaces to learn compressed representations, thereby streamlining both training and inference (Rombach et al., 2022; Hu et al., 2024b). This latent learning approach integrates naturally with modern neural architectures and can be extended to various modalities beyond images, showcasing the versatility of diffusion processes in modeling complex data distributions. In parallel, recent researchers have also explored multi-scale noise scheduling and adaptive step-size strategies to enhance convergence stability and maintain high-resolution detail in generated content in (Lovelace et al., 2024; Feng et al., 2024; Rout et al., 2024; Jiang et al., 2025; Luo et al., 2024).

**Flow Matching.** Generative models like diffusion (Sohl-Dickstein et al., 2015; Ho et al., 2020; Song et al., 2020) and flow-matching (Lipman et al., 2022; Liu et al., 2022a) operate by learning ordinary differential equations (ODEs) that map noise to data. To simplify, this study leverages the optimal transport flow-matching formulation (Liu et al., 2022a). A linear combination of a noise sample $x_0 \sim \mathcal{N}(0, \mathbb{I})$ and a data point $x_1 \sim \mathcal{D}$ defines $x_t = (1-t)x_0 + tx_1, v_t = x_1 - x_0$, with $v_t$ representing the velocity vector directed from $x_0$ to $x_1$. While $v_t$ is uniquely derived from $(x_0, x_1)$, knowledge of only $x_t$ renders it a random variable due to the ambiguity in selecting $(x_0, x_1)$. Neural networks in flow models approximate the expected velocity $\bar{v}_t = \mathbb{E}[v_t \mid x_t]$, calculated as an average over all valid pairings. Training involves minimizing the deviation between predicted and empirical

velocities:

$$\bar{v}_\theta(x_t, t) \sim \mathbb{E}_{x_0, x_1 \sim \mathcal{D}} \left[ v_t \mid x_t \right]$$
$$\mathcal{L}^{\mathrm{F}}(\theta) = \mathbb{E}_{x_0, x_1 \sim \mathcal{D}} \left[ \| \bar{v}_\theta(x_t, t) - (x_1 - x_0) \|^2 \right]. \tag{1}$$

Sampling involves first drawing a noise point $x_0 \sim \mathcal{N}(0, I)$ and iteratively transforming it into a data point $x_1$. The denoising ODEs, parameterized by $\bar{v}_\theta(x_t, t)$, governs this transformation, and Euler's method approximates it over small, discrete time steps.

**High-order ODE Gradient in Diffusion Models.** Higher-order gradient-based methods like TTMs (Kloeden & Platen, 1992) have applications far exceeding DDMs. For instance, solvers (Djeumou et al., 2022) and regularization frameworks (Kelly et al., 2020; Finlay et al., 2020) for neural ODEs (Chen et al., 2018; Grathwohl et al., 2018) frequently utilize higher-order derivatives. Beyond machine learning contexts, the study of higher-order TTMs has been extensively directed toward solving stiff (Chang & Corliss, 1994) and non-stiff (Chang & Corliss, 1994; Corliss & Chang, 1982) systems.

## 3  PRELIMINARY

This section elaborates on the flow-matching framework, extending it to the second-order case, with critical definitions underscored. We begin with describing the general framework of flow matching and its second-order rectification. These concepts form the basis for our proposed method, as they integrate first and second-order information for trajectory estimation.

**Fact 3.1.** *Let a field $x_t$ be defined as $x_t = \alpha_t x_0 + \beta_t x_1$, where $\alpha_t$ and $\beta_t$ are functions of $t$, and $x_0, x_1$ are constants. Then, the first-order gradient $\dot{x}_t$ and the second-order gradient $\ddot{x}_t$ can be manually calculated as $\dot{x}_t = \dot{\alpha}_t x_0 + \dot{\beta}_t x_1$, $\ddot{x}_t = \ddot{\alpha}_t x_0 + \ddot{\beta}_t x_1$., reprectively.*

In practice, one often samples $(x_0, x_1)$ from $(\mu_0, \pi_0)$ and parameterizes $x_t$ (e.g., interpolation) at intermediate times to build a training objective that matches the velocity field to the true time derivative $\dot{x}_t$.

**Definition 3.2** (Shortcut models, implicit definition from page 3 on (Frans et al., 2025))**.** *Let $\Delta t = 1/128$. Let $x_t$ be current field. Let $t \in \mathbb{N}$ denote time step. Let $u_1(x_t, t, d)$ be the network to be trained. Let $d \in (1/128, 1/64, \ldots, 1/2, 1)$ denote step size. Then, we define Shortcut model compute next field $x_{t+d}$ as follow:*

$$x_{t+d} = \begin{cases} x_t + u_1(x_t, t, d)d & \text{if } d \geq 1/128, \\ x_t + u_1(x_t, t, 0)\Delta t & \text{if } d < 1/128. \end{cases}$$

## 4  METHODOLOGY

When training a flow-based model, such as Shortcut model, using only the first-order term as the training loss has several limitations compared to incorporating high-order losses. (1) Firstly, relying solely on the first-order term results in a less accurate approximation of the true dynamics, as it captures only the linear component and misses important nonlinear aspects that higher-order terms can represent. This can lead to slower convergence, as the model must implicitly learn complex dynamics without explicit guidance from higher-order terms. (2) Additionally, while the first-order approach reduces model complexity and the risk of overfitting, it may also limit the model's ability to generalize effectively to unseen data, particularly when the underlying dynamics are highly nonlinear. (3) In contrast, including higher-order terms enhances the model's capacity to capture intricate patterns, improving both accuracy and generalization, albeit at the cost of increased computational complexity and potential overfitting risks.

Then, we introducing our HOMO model. The intuition behind this design is to leverage high-order dynamics to achieve a more accurate and stable approximation of the field evolution. By incorporating higher-order losses, we aim to capture the nonlinearities and complex interactions that are often present in real-world systems. This approach not only improves the fidelity of the model but also enhances its ability to generalize across different scenarios.

**Definition 4.1** (HOMO Inference). *Let $\Delta t = 1/128$. Let $x_t$ be the current field. Let $t \in \mathbb{N}$ denote the time step. Let $u_{1,\theta_1}(\cdot)$ and $u_{2,\theta_2}(\cdot)$ denote the HOMO models to be trained. Let $d \in (0, 1/128, 1/64, \ldots, 1/2, 1)$ denote the step size. Then, we define the HOMO computation of the next field $x_{t+d}$ as follows:*

$$x_{t+d} = \begin{cases} x_t + d \cdot u_1(x_t, t, d) + \frac{d^2}{2} \cdot u_2(u_1(x_t, t, d), x_t, t, d) & \text{if } d \geq 1/128, \\ x_t + \Delta t \cdot u_1(x_t, t, 0) + \frac{(\Delta t)^2}{2} \cdot u_2(u_1(x_t, t, 0), x_t, t, 0) & \text{if } d < 1/128. \end{cases}$$

The self-consistency target is to ensure that the model's predictions are consistent across different time steps, which is crucial for maintaining the stability and accuracy of the model over long-term predictions.

**Definition 4.2** (HOMO Self-Consistency Target). *Let $u_{1,\theta_1}$ be the networks to be trained. Let $x_t$ be the current field and $x_{t+d}$ be defined in Definition 4.1. Let $t \in \mathbb{N}$ denote the time step. Let $d \in (0, 1/128, 1/64, \ldots, 1/2, 1)$ denote the step size. We define the Self-Consistency target as follows:*

$$\dot{x}_t^{\text{target}} = u_{1,\theta_1}(x_t, t, d)/2 + u_{1,\theta_1}(x_{t+d}, t, d)/2$$

The second-order HOMO loss is designed to optimize the model by minimizing the discrepancy between the predicted and true velocities and accelerations. This loss function ensures that the model not only captures the immediate dynamics but also the underlying trends and changes in the system.

**Definition 4.3** (Second-order HOMO Loss). *Let $x_t$ be the current field. Let $t \in \mathbb{N}$ denote the time step. Let $\dot{x}_t^{\text{target}}$ be defined by Definition 4.2. Let $u_{1,\theta_1}(\cdot)$ and $u_{2,\theta_2}(\cdot)$ denote the HOMO models to be trained. Let $d \in (0, 1/128, 1/64, \ldots, 1/2, 1)$ denote the step size. Let $\dot{x}_t^{\text{true}}$ and $\ddot{x}_t^{\text{true}}$ be the observed (or numerically approximated) true velocity and acceleration. Let $\dot{x}_t^{\text{pred}} := u_{1,\theta_1}(x_t, t, 2d)$ denote the model prediction of the first-order term. Then, we define the HOMO Loss as follows:*

$$L_{(\theta_1, \theta_2)} = \mathbb{E}[\ell_{2,1,\theta_1}(x_t, \dot{x}_t^{\text{true}})] + \mathbb{E}[\ell_{2,2,\theta_2,\theta_1}(x_t, \ddot{x}_t^{\text{true}})] + \mathbb{E}[\|u_{1,\theta_1}(x_t, t, 2d) - \dot{x}_t^{\text{target}}\|^2]$$

*We define*

$$\ell_{2,1,\theta_1}(x_t, \dot{x}_t^{\text{true}}) := \|u_{1,\theta_1}(x_t, t, 2d) - \dot{x}_t^{\text{true}}\|^2,$$
$$\ell_{2,2,\theta_2,\theta_1}(x_t, \ddot{x}_t^{\text{true}}) := \|u_{2,\theta_2}(\dot{x}_t^{\text{pred}}, x_t, t, 2d) - \ddot{x}_t^{\text{true}}\|^2$$
$$\ell_{\text{selfc}}(x_t, \dot{x}_t^{\text{target}}) := \|u_{1,\theta_1}(x_t, t, 2d) - \dot{x}_t^{\text{target}}\|^2$$

*and*

$$\ell_{(\theta_1, \theta_2)}(x_t, x_t^{\text{true}}) := \ell_{2,1,\theta_1}(x_t, \dot{x}_t^{\text{true}}) + \ell_{2,2,\theta_2,\theta_1}(x_t, \ddot{x}_t^{\text{true}}) + \ell_{\text{selfc}}(x_t, \dot{x}_t^{\text{target}}).$$

**Remark 4.4** (Simple notations). *For simplicity, we denote first-order matching as M1, which implies that HOMO is optimized solely by the first-order loss $\ell_{2,1,\theta_1}(x_t, \dot{x}_t^{\text{true}})$. Second-order matching is denoted as M2, where HOMO is optimized only by the second-order loss $\ell_{2,2,\theta_2,\theta_1}(x_t, \ddot{x}_t^{\text{true}})$. We refer to HOMO optimized solely by the self-consistency loss as SC, denoted by $\ell_{\text{selfc}}(x_t, \dot{x}_t^{\text{target}})$. Combinations of M1, M2, and SC are used to indicate HOMO optimized by corresponding combinations of loss terms. For example, (M1 + M2) denotes HOMO optimized by both first-order and second-order terms, while (M1 + M2 + SC) represents HOMO optimized by the first-order, second-order, and self-consistency terms.*

## 5 THEORETICAL ANALYSIS

In this section, we will introduce our main result, the approximation error of the second order flow matching. The theory for higher order flow matching is deferred to Section D.

We first present the approximation error result for the early stage of the diffusion process. This result establishes theoretical guarantees on how well a neural network can approximate the first and second order flows during the initial phases of the trajectory evolution.

---

**Algorithm 1** HOMO Training

---

1: **procedure** HOMOTRAINING($\theta, D, p, k$)
2:                                                          ▷ Parameter $\theta$ for HOMO model $u_1$ and $u_2$.
3:                                     ▷ Training dataset $D$
4:                                   ▷ Stepsize and time index distribution $p$
5:                                   ▷ Batch size $k$
6:      **while** not converged **do**
7:          $x_0 \sim \mathcal{N}(0, I), x_1 \sim D, (d, t) \sim p$
8:          $\beta_t \leftarrow \sqrt{1 - \alpha_t^2}$
9:          $x_t \leftarrow \alpha_t \cdot x_0 + \beta_t \cdot x_1$                      ▷ Noise data point
10:          **for** first $k$ batch elements **do**
11:              $\dot{s}_t^{\text{true}} \leftarrow \dot{\alpha}_t x_0 + \dot{\beta}_t x_1$                ▷ First-order target
12:              $\ddot{s}_t^{\text{true}} \leftarrow \ddot{\alpha}_t x_0 + \ddot{\beta}_t x_1$              ▷ Second-order target
13:              $d \leftarrow 0$
14:          **end for**
15:          **for** other batch elements **do**
16:              $s_t \leftarrow u_1(x_t, t, d)$                ▷ First small step of first order
17:              $\dot{s}_t \leftarrow u_2(u_1(x_t, t, d), x_t, t, d)$       ▷ First small step of second order
18:              $x_{t+d} \leftarrow x_t + d \cdot s_t + \frac{d^2}{2} \dot{s}_t$           ▷ Follow ODE
19:              $s_{t+d} \leftarrow u_1(x_{t+d}, t + d, d)$         ▷ Second small step of first order
20:              $\dot{s}_t^{\text{target}} \leftarrow \text{stopgrad} (s_t + s_{t+d})/2$     ▷ Self-consistency target of first order
21:          **end for**
22:          $\theta \leftarrow \nabla_\theta(\|u_1(x_t, t, 2d) - \dot{s}_t^{\text{true}}\|^2$
                   $+\|u_2(u_1(x_t, t, 2d), x_t, t, 2d) - \ddot{s}_t^{\text{true}}\|^2$
                   $+\|u_1(x_t, t, 2d) - \dot{s}_t^{\text{target}}\|^2)$
23:      **end while**
24:      **return** $\theta$
25: **end procedure**

---

**Algorithm 2** HOMO Sampling

---

1: **procedure** HOMOSAMPLING($\theta, M$)
2:                                   ▷ Parameter $\theta$ for the HOMO model $u_1$ and $u_2$
3:                                   ▷ The number of sampling steps $M$
4:      $x \sim \mathcal{N}(0, I)$
5:      $d \leftarrow 1/M$
6:      $t \leftarrow 0$
7:      **for** $n \in [0, \ldots, M - 1]$ **do**
8:          $x \leftarrow x + d \cdot u_1(x, t, d) + \frac{d^2}{2} \cdot u_2(u_1(x, t, d), x, t, d)$
9:          $t \leftarrow t + d$
10:      **end for**
11:      **return** $x$
12: **end procedure**

---

**Theorem 5.1** (Approximation error of second order flow matching for small $t$, informal version of Theorem D.1). *Let $N$ be a value associated with sample size $n$. Let $T_0 := N^{-R_0}$ and $T_* := N^{-\frac{\kappa^{-1} - \delta}{d}}$ where $R_0, \kappa, \delta$ are some parameters. Let $s$ be the order of smoothness of the Besov space that the target distribution belongs to. Under some mild assumptions, there exist neural networks $\phi_1, \phi_2$ from a class of neural networks such that, for sufficiently large $N$, we have $\int (\|\phi_1(x, t) - \dot{x}_t^{\text{true}}\|_2^2 + \|\phi_2(x, t) - \ddot{x}_t^{\text{true}}\|_2^2) p_t(x) \mathrm{d}x \lesssim (\dot{\alpha}_t^2 \log N + \dot{\beta}_t^2) N^{-\frac{2s}{d}} + \mathbb{E}_{x_t \sim P_t}[\|\dot{x}_t^{\text{true}} - \ddot{x}_t^{\text{true}}\|_2^2]$ holds for any $t \in [T_0, 3T_*]$. In addition, $\phi_1, \phi_2$ can be taken so we have $\|\phi_1(\cdot, t)\|_\infty = O(|\dot{\alpha}_t| \sqrt{\log n} + |\dot{\beta}_t|), \|\phi_2(\cdot, t)\|_\infty = O(|\dot{\alpha}_t| \sqrt{\log n} + |\dot{\beta}_t|).$*

Next, we present the approximation error result for the later stages, confirming that the second-order flow matching remains effective throughout the generative process.

**Theorem 5.2** (Approximation error of second order flow matching for large $t$, informal version of Theorem D.3). *Let $N$ be a value associated with sample size $n$. Let $T_0 := N^{-R_0}$ and $T_* := N - \frac{\kappa^{-1}-\delta}{d}$ where $R_0, \kappa, \delta$ are some parameters. Let $s$ be the order of smoothness of the Besov space that the target distribution belongs to. Fix $t_* \in [T_*, 1]$ and let $\eta > 0$ be arbitrary. Under some mild assumptions, there exists neural networks $\phi_1, \phi_2$ from a class of neural networks such that $\int(\|\phi_1(x,t) - \dot{x}_t^{\text{true}}\|_2^2 + \|\phi_2(x,t) - \ddot{x}_t^{\text{true}}\|_2^2)p_t(x)\mathrm{d}x \lesssim (\dot{\alpha}_t^2 \log N + \dot{\beta}_t^2)N^{-\eta} + \mathbb{E}_{x_t \sim P_t}[\|\dot{x}_t^{\text{true}} - \ddot{x}_t^{\text{true}}\|_2^2]$ holds for any $t \in [2t_*, 1]$. In addition, $\phi_1, \phi_2$ can be taken so we have*

$$\|\phi_1(\cdot, t)\|_\infty = O(|\dot{\alpha}_t| \log N + |\dot{\beta}_t|),$$
$$\|\phi_2(\cdot, t)\|_\infty = O(|\dot{\alpha}_t| \log N + |\dot{\beta}_t|).$$

Overall, these two results demonstrate the effectiveness across different phases of the generative process.

## 6 EXPERIMENTS

This section presents a series of experiments to evaluate the effectiveness of our HOMO method and assess the impact of each loss component. Our results demonstrate that HOMO significantly improves distribution generation, with the high-order loss playing a key role in enhancing model performance.

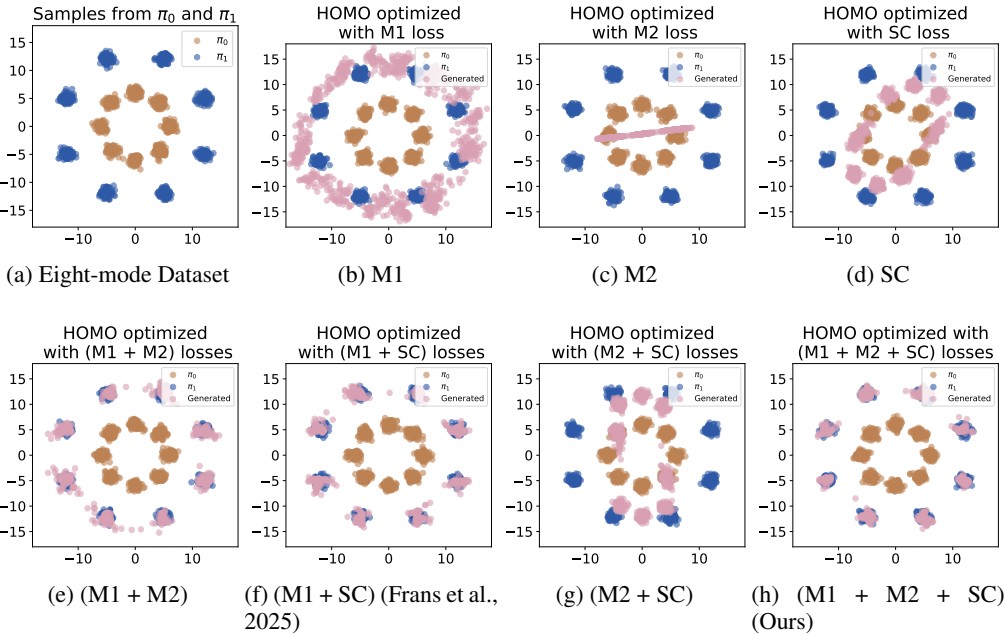

Figure 1: The first row shows the initial eight-mode dataset (a) and the results of HOMO optimized with first-order loss (M1), second-order loss (M2), and self-consistency loss (SC) Figures (b-d). The second row presents combinations of losses: M1+M2 (e), M1+SC (Frans et al., 2025) (f), M2+SC (g), and M1+M2+SC (Ours) (h). Quantitative results are shown in Table 1

### 6.1 EXPERIMENT SETUP

We evaluate HOMO on a variety of data distributions and different combinations of losses. We would like to restate that the HOMO with first order loss and self-consistency loss is equal to the original One-step Shortcut model (Frans et al., 2025), i.e., M1+SC. Furthermore, M1+M2+SC and M1+M2+M3+SC are our proposed methods.

Table 1: **Euclidean distance loss on Gaussian datasets.** Lower values indicate more accurate distribution matching. Optimal values are in **Bold**, with Underlined numbers representing second-best results. For qualitative results, please refer to Figure 1.

| Losses | Four mode ($\downarrow$) | Five mode ($\downarrow$) | Eight mode ($\downarrow$) |
|---|---|---|---|
| M1 | 2.759 | 3.281 | 3.321 |
| M2 | 11.089 | 6.554 | 10.830 |
| SC | 6.761 | 10.893 | 7.646 |
| M1 + M2 | 0.941 | 1.097 | 0.977 |
| M2 + SC | 8.708 | 9.212 | 4.801 |
| M1 + SC (Frans et al., 2025) | 0.820 | 1.067 | 1.084 |
| M1 + M2 + SC (Ours) | **0.809** | **0.917** | **0.778** |

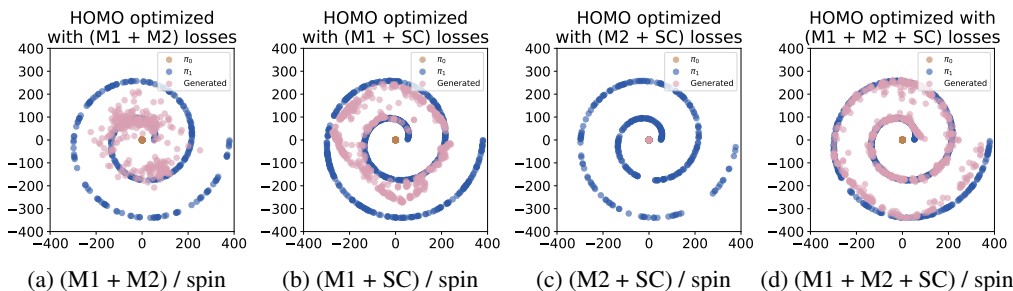

(a) (M1 + M2) / spin    (b) (M1 + SC) / spin    (c) (M2 + SC) / spin    (d) (M1 + M2 + SC) / spin

Figure 2: **HOMO on complex datasets (Spin).** Results of HOMO optimized with various loss combinations: M1+M2 (a), M1+SC (Frans et al., 2025) (b), M2+SC (c), and M1+M2+SC (Ours) (d). Quantitative results are in Table 2.

Table 2: **Euclidean distance loss on complex datasets.** Lower values indicate better distribution matching. For qualitative results of complex distribution experiments, please refer to Figure 2 and Figure 13, 14, 15, 16.

| Losses | Circle ($\downarrow$) | Irregular ($\downarrow$) | Spiral ($\downarrow$) | Spin ($\downarrow$) |
|---|---|---|---|---|
| M1 + M2 | 0.642 | 0.731 | 7.233 | 31.009 |
| M1 + SC | 0.736 | 0.743 | 3.289 | 12.055 |
| M2 + SC | 7.233 | 0.975 | 10.096 | 50.499 |
| M1 + M2 + SC | **0.579** | **0.678** | **1.840** | **10.066** |

For the distribution dataset, in the left-most figure of Figure 1, we show an example of an eight-mode Gaussian distribution. The source distribution $\pi_0$ and the target distribution $\pi_1$ are constructed as mixture distributions, each consisting of eight equally weighted Gaussian components. Each Gaussian component has a variance of $0.3$. This setup presents a challenging transportation problem, requiring the flow to handle multiple modes and cross-modal interactions.

We implement HOMO according to three kinds of losses: the first-order loss $\ell_{2,1,\theta_1}(x_t, \dot{x}_t^{\mathrm{true}})$, the second-order loss $\ell_{2,2,\theta_2,\theta_1}(x_t, \ddot{x}_t^{\mathrm{true}})$, and the self-consistency loss $\ell_{\mathrm{selfc}}(x_t, \dot{x}_t^{\mathrm{target}})$. Following Remark 4.4, we use M1, M2 and SC to denote three kinds of losses, respectively. For all experiments, we optimize models by the sum of squared error. For the target transport trajectory setting, we follow the VP ODE framework from (Liu et al., 2022b), which is $x_t = \alpha_t x_0 + \beta_t x_1$. We choose $\alpha_t = \exp(-\frac{1}{4}a(1-t)^2 - \frac{1}{2}b(1-t))$ and $\beta_t = \sqrt{1 - \alpha_t^2}$, with $a = 19.9$ and $b = 0.1$.

## 6.2 MIXTURE OF GAUSSIAN EXPERIMENTS

We analyze the performance of HOMO on Gaussian mixture datasets (Liang et al., 2024e) with varying modes (four, five, and eight). The most challenging is the eight-mode distribution, where

HOMO with all three losses (M1+M2+SC) produces the best results, achieving the lowest Euclidean distance. The eight-mode Gaussian mixture distribution dataset (Figure 1 (a) ) contains eight Gaussian distributions whose variance is $0.3$. Eight source mode (**brown**) positioned at a distance $D_0 = 6$ from the origin, and eight target mode (**indigo**) positioned at a distance $D_0 = 13$ from the origin, each mode sample 100 points. HOMO optimized with M1+M2+SC losses is the only model that can accurately learn the target eight-mode Gaussian distribution, achieving high precision as evidenced by the lowest Euclidean distance loss among all tested configurations. We emphasize the importance of the second-order loss. Without it, the model struggles to accurately capture finer distribution details (Figure 1 (f)). However, when included, the model better matches the target distribution (Figure 1 (h)).

We further analyze how each loss contributes to the final performance of the HOMO. (1) The first-order loss enables HOMO to learn the general structure of the target distribution, but it struggles to capture finer details, as shown in Figure 1 (b) and Figure 1 (g). (2) The second-order loss can lead to overfitting in the target distribution, as shown in Figure 1(c). When used alone, the second-order loss may cause the model to focus too much on details and lose sight of the broader distribution. (3) The self-consistency loss enhances the concentration of the learned distribution, as shown in Figure 1 (d). Without the self-consistency loss, as shown in Figure 1 (e), the learned distribution becomes sparser.

## 6.3 COMPLEX DISTRIBUTION EXPERIMENTS

In this section, we conduct experiments on datasets with complex distributions, where we expect our HOMO model to learn the transformation from a regular source distribution to an irregular target distribution. We first introduce the dataset used in Figure 2. In the spin dataset, we sample 600 points from a Gaussian distribution with a variance of $0.3$ for both the source and target distributions. The second-order loss is essential for accurate fitting, particularly for irregular and spiral distributions. We emphasize the critical role of the second order loss in the success of our HOMO model for learning complex distributions. As demonstrated in Figure 2 (b), the original shortcut model, which includes only first-order and self-consistency losses, fails to accurately fit the outer circle distribution. In contrast, the result of our HOMO model, shown in Figure 2 (d), illustrates that adding the second-order loss enables the model to generate points within the outer circle. This highlights the importance of the second-order loss in enabling the model to learn more complex distributions. We have also performed experiments with HOMO optimized using each loss individually, as well as on other distribution datasets. Due to space limitations, we refer the reader to Section E.2, E.3, and E.4 for further details.

## 6.4 THIRD-ORDER HOMO

Table 3: **Euclidean distance loss of three complex distribution datasets under original trajectory setting.** Lower values indicate more accurate distribution transfer results. For the qualitative results , please refer to Figure 3.

| Loss terms | 2 Round Spin ($\downarrow$) | 3 Round Spin ($\downarrow$) | Dot-Circle ($\downarrow$) |
|---|---|---|---|
| SC | 59.490 | 50.981 | 89.974 |
| M1 + SC (Frans et al., 2025) | 17.866 | 23.606 | 37.550 |
| M1 + M2 + SC (Ours) | 9.417 | 13.085 | 30.679 |
| M1 + M2 + SC + M3 (Ours) | **7.440** | **10.679** | **26.819** |

As discussed in previous sections, second-order HOMO has shown great performance on various distribution datasets. Therefore, in this section, we further investigate the performance of HOMO for adding an additional third-order loss. We use the same datasets as discussed in previous sections. Namely, they are 2 Round spin, 3 Round spin, and Dot-Circle datasets. The qualitative results from the experiments (Figure 3) demonstrate that the additional third-order loss enables HOMO to better capture more complex target distributions. For instance, the comparison between Figure 3 (c) and Figure 3 (d), as well as between Figure 3 (g) and Figure 3 (h), illustrates how the third-order loss enhances the model's ability to fit more intricate distributions. These findings are consistent with the quantitative results presented in Table 3. By capturing higher-order interactions, the model is

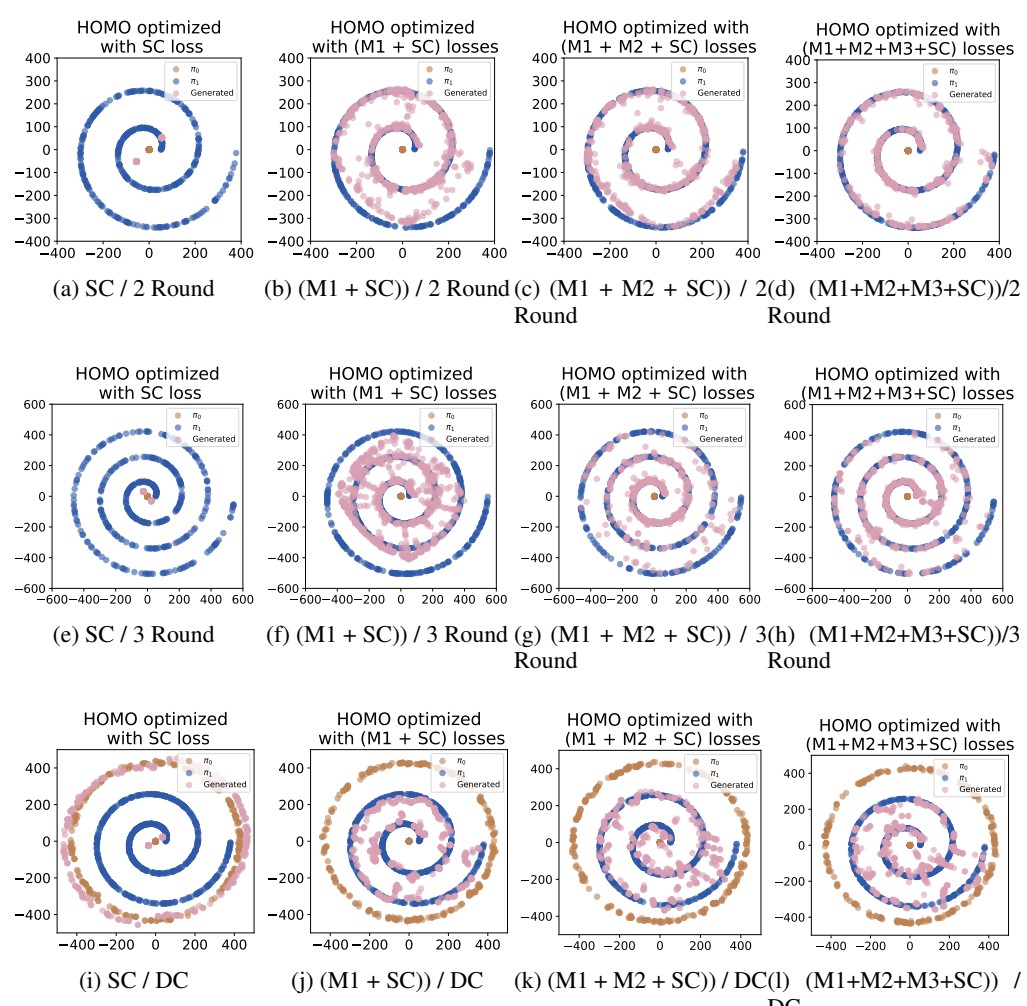

Figure 3: We present the third-order HOMO results in three kinds of complex datasets. From left to right, we present results of HOMO optimized with different kinds of losses. A quantitative evaluation is presented in Table 3.

better equipped to understand and adapt to the intricate relationships within the data, leading to more accurate and robust learning. This approach underscores the value of enriching the model's training objective to handle the complexities inherent in real-world data distributions.

# 7 CONCLUSION

In this work, we introduced HOMO , which incorporates high-order dynamics into the training and sampling processes of Shortcut models. By leveraging high-order supervision, our method significantly enhances the geometric consistency and precision of learned trajectories. Theoretical analyses demonstrate that high-order supervision ensures stability and generalization across different phases of the generative process. These findings are supported by extensive experiments, where HOMO outperforms original Shortcut models (Frans et al., 2025), achieving more accurate distributional alignment and fewer suboptimal trajectories. The integration of high-order terms establishes a new style for geometrically-aware generative modeling, highlighting the importance of capturing higher-order dynamics for accurate transport learning. Our results suggest that high-order supervision is a powerful tool for improving the fidelity and robustness of generative models.

## ETHIC STATEMENT

This paper does not involve human subjects, personally identifiable data, or sensitive applications. We do not foresee direct ethical risks. We follow the ICLR Code of Ethics and affirm that all aspects of this research comply with the principles of fairness, transparency, and integrity.

## REPRODUCIBILITY STATEMENT

We ensure reproducibility on both theoretical and empirical fronts. For theory, we include all formal assumptions, definitions, and complete proofs in the appendix. For experiments, we describe model architectures, datasets, preprocessing steps, hyperparameters, and training details in the main text and appendix.

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

# Appendix

**Roadmap.**    In Section A, we introduce the Shortcut Model Training and Sampling Algorithm. Section B discusses related works that inspire our approach. Section C states the tools from (Fukumizu et al., 2024) used in our analysis. Section D explores the theory behind Higher-Order Flow Matching. Section E investigates the impact of different optimization terms through empirical ablation studies. Section F examines model performance on complex distribution experiments. Section G extends HOMO to third-order dynamics and evaluates its effectiveness on complex tasks. Section H quantifies the computational and optimization costs associated with different configurations.

## A    ORIGINAL ALGORITHM

Here we introduce Shortcut Model Training and Sampling Algorithm from Page 5 of (Frans et al., 2025)

---

**Algorithm 3** Shortcut Model Training from page 5 of (Frans et al., 2025)

---

1: **while** not converged **do**
2:     $x_0 \sim \mathcal{N}(0, I), x_1 \sim D, (d, t) \sim p(d, t)$
3:     $x_t \leftarrow (1 - t)x_0 + tx_1$                              ▷ Noise data point
4:     **for** first $k$ batch elements **do**
5:         $s_{\text{target}} \leftarrow x_1 - x_0$                       ▷ Flow-matching target
6:         $d \leftarrow 0$
7:     **end for**
8:     **for** other batch elements **do**
9:         $s_t \leftarrow u_1(x_t, t, d)$                              ▷ Fitst small step
10:         $x_{t+d} \leftarrow x_t + s_t d$                            ▷ Follow ODE
11:         $s_{t+d} \leftarrow u_1(x_{t+d}, t + d, d)$                   ▷ Second small step
12:         $s_{\text{target}} \leftarrow \text{stopgrad} \, (s_t + s_{t+d})/2$   ▷ Self-consistency target
13:     **end for**
14:     $\theta \leftarrow \nabla_\theta \|u_1(x_t, t, 2d) - s_{\text{target}}\|^2$
15: **end while**

---

**Algorithm 4** Shortcut model. Sampling from page 5 of (Frans et al., 2025)

---

1: $x \sim \mathcal{N}(0, I)$
2: $d \leftarrow 1/M$
3: $t \leftarrow 0$
4: **for** $n \in [0, \dots, M - 1]$ **do**
5:     $x \leftarrow x + d \cdot u_1(x, t, d)$
6:     $t \leftarrow t + d$
7: **end for**
8: **return** $x$

---

## B    MORE RELATED WORK

In this section, we discuss more related work that inspires our work.

**Large Language Models.**    Neural networks built upon the Transformer architecture (Vaswani et al., 2017) have swiftly risen to dominate modern machine learning approaches in natural language processing. Extensive Transformer models, trained on wide-ranging and voluminous datasets while encompassing billions of parameters, are often termed large language models (LLM) or foundation models (Bommasani et al., 2021). Representative instances include BERT (Devlin et al., 2019), PaLM (Chowdhery et al., 2022), Llama (Touvron et al., 2023), ChatGPT (OpenAI, 2024), GPT4 (OpenAI, 2023), among others. These LLMs have showcased striking general intelligence abilities (Bubeck et al., 2023) in various downstream tasks. Numerous adaptation methods have

been developed to tailor LLMs for specific applications, such as adapters (Hu et al., 2022; Zhang et al., 2023; Gao et al., 2023a; Shi et al., 2023), calibration schemes (Zhao et al., 2021; Zhou et al., 2023), multitask fine-tuning (Gao et al., 2021a; Xu et al., 2023; Von Oswald et al., 2023; Xu et al., 2024), prompt optimization (Gao et al., 2021b; Lester et al., 2021), scratchpad approaches (Nye et al., 2021), instruction tuning (Li & Liang, 2021; Chung et al., 2022; Mishra et al., 2022), symbol tuning (Wei et al., 2023), black-box tuning (Sun et al., 2022), and reinforcement learning from human feedback (RLHF) (Ouyang et al., 2022). Additional lines of research endeavor to boost model efficiency without sacrificing performance across diverse domains, for example in (Deng et al., 2022; Song et al., 2023; Gao et al., 2023c;e;d; Bian et al., 2023; Deng et al., 2023; Gao et al., 2023b; Shrivastava et al., 2023; Qin et al., 2023; Chen et al., 2024c; Li et al., 2024d; Chen et al., 2024b; Liang et al., 2024c; Chen et al., 2024a; Liang et al., 2024b;d;a; Li et al., 2024a;c; Cao et al., 2024; Li et al., 2024b; Chen et al., 2024e;d; Ke et al., 2024; 2025a;b; Li et al., 2025; Hu et al., 2024a).

## C  TOOLS FROM PREVIOUS WORKS

We state the tools in (Fukumizu et al., 2024) that we will use to prove our main results.

### C.1  DEFINITIONS OF BESOV SPACE

**Definition C.1** (Modulus of Smoothness). *Let $\Omega$ be a domain in $\mathbb{R}^d$. For a function $f \in L^{p'}(\Omega)$ with $p' \in (0, \infty]$, the $r$-th modulus of smoothness of $f$ is defined by*

$$w_{r,p'}(f, t) = \sup_{\|h\|_2 \leq t} \|\Delta_h^r(f)\|_{p'},$$

*where the finite difference operator $\Delta_h^r(f)(x)$ is given by*

$$\Delta_h^r(f)(x) = \begin{cases} \sum_{j=0}^{r} \binom{r}{j}(-1)^{r-j} f(x + jh), & \text{if } x + jh \in \Omega \text{ for all } j, \\ 0, & \text{otherwise.} \end{cases}$$

**Definition C.2** (Besov Seminorm). *Let $0 < p', q' \leq \infty$, $s > 0$, and set $r := |s| + 1$. The Besov seminorm of $f \in L^{p'}(\Omega)$ is defined as*

$$|f|_{B_{p',q'}^s} := \begin{cases} \left( \int_0^\infty (t^{-s} w_{r,p'}(f, t))^{q'} \frac{dt}{t} \right)^{\frac{1}{q'}}, & q' < \infty, \\ \sup_{t>0} t^{-s} w_{r,p'}(f, t), & q' = \infty. \end{cases}$$

**Definition C.3** (Besov Space). *The Besov space $B_{p',q'}^s(\Omega)$ is the function space equipped with the norm*

$$\|f\|_{B_{p',q'}^s} := \|f\|_{p'} + |f|_{B_{p',q'}^s},$$

*It consists of all functions $f \in L^{p'}(\Omega)$ such that*

$$B_{p',q'}^s(\Omega) := \{ f \in L^{p'}(\Omega) \mid \|f\|_{B_{p',q'}^s} < \infty \}.$$

**Remark C.4.** *The parameter $s$ governs the degree of smoothness of functions in $B_{p',q'}^s(\Omega)$. In particular, when $p' = q'$ and $s$ is an integer, the Besov space $B_{p',q'}^s(\Omega)$ coincides with the standard Sobolev space of order $s$. For further details on the properties and applications of Besov spaces, see (Triebel, 1992).*

### C.2  B-SPLINE

**Definition C.5** (Indicator Function). *Let $\mathcal{N}(x)$ be the characteristic function defined by*

$$\mathcal{N}(x) = \begin{cases} 1, & x \in [0, 1], \\ 0, & \text{otherwise.} \end{cases}$$

**Definition C.6** (Cardinal B-Spline). *For $\ell \in \mathcal{N}$, the cardinal B-spline of order $\ell$ is defined by*

$$\mathcal{N}_\ell(x) := \underbrace{\mathcal{N} * \mathcal{N} * \cdots * \mathcal{N}}_{\ell + 1 \text{ times}}(x),$$

where $*$ denotes the convolution operation. Explicitly, the convolution of two functions $f, g : \mathbb{R} \to \mathbb{R}$ is given by

$$(f * g)(x) = \int_{\mathbb{R}} f(x - y)g(y)\mathrm{d}y.$$

Thus, $\mathcal{N}_\ell(x)$ is obtained by convolving $\mathcal{N}$ with itself $(\ell + 1)$ times.

**Definition C.7** (Tensor Product B-Spline Basis). *For a multi-index $k \in \mathcal{N}^d$ and $j \in \mathbb{Z}^d$, the tensor product B-spline basis in $\mathbb{R}^d$ of order $\ell$ is defined as*

$$M_{k,j}^d(x) := \prod_{i=1}^{d} \mathcal{N}_\ell(2^{k_i} x_i - j_i).$$

*This basis is constructed as the product of univariate B-splines, scaled and translated according to the parameters $k$ and $j$.*

**Definition C.8** (B-Spline Approximation in Besov Spaces in (Suzuki, 2019; Oko et al., 2023)). *A function $f$ in the Besov space can be approximated using a superposition of tensor product B-splines as*

$$f_N(x) = \sum_{(k,j)} \alpha_{k,j} M_{k,j}^d(x),$$

*where the summation is taken over appropriate index sets $(k, j)$, and the coefficients $\alpha_{k,j}$ are real numbers that determine the contribution of each basis function.*

## C.3 CLASS OF NEURAL NETWORKS

**Definition C.9** (Neural Network Class in (Fukumizu et al., 2024)). *Let $L \in \mathbb{N}$ denote the depth (number of layers), $W = (W_1, W_2, \ldots, W_{L+1}) \in \mathbb{N}^{L+1}$ the width configuration of the network, $S \in \mathbb{N}$ a sparsity constraint, and $B > 0$ a norm bound. The class of neural networks $\mathcal{M}(L, W, S, B)$ is defined as*

$$\mathcal{M}(L, W, S, B) := \{ \psi_{A^{(L)}, b^{(L)}} \circ \cdots \circ \psi_{A^{(2)}, b^{(2)}}(A^{(1)}x + b^{(1)})m | A^{(i)} \in \mathbb{R}^{W_{i+1} \times W_i}, b^{(i)} \in \mathbb{R}^{W_{i+1}},$$

$$\sum_{i=1}^{L}(\|A^{(i)}\|_0 + \|b^{(i)}\|_0) \le S, \quad \max_{1 \le i \le L}\{\|A^{(i)}\|_\infty \vee \|b^{(i)}\|_\infty\} \le B\}.$$

*Here, the function $\psi_{A,b} : \mathbb{R}^{W_i} \to \mathbb{R}^{W_{i+1}}$ represents the affine transformation with ReLU activation, given by*

$$\psi_{A,b}(z) = A \cdot \mathsf{ReLU}(z) + b, \quad \text{where} \quad \mathsf{ReLU}(z) = \max\{0, z\}.$$

*The sparsity constraint ensures that the total number of nonzero entries in all weight matrices and bias vectors does not exceed $S$, while the norm constraint limits their maximum absolute values to $B$.*

## C.4 ASSUMPTIONS

**Remark C.10.** *We introduce a small positive constant $\delta > 0$ and denote by $N$ the number of basis functions in the B-spline used to approximate $p_t(x)$. The value of $N$ is determined by the sample size $n$, specifically following the relation $N = n^{\frac{d}{2s+d}}$, which balances the approximation error and the complexity of both the B-spline and the neural network.*

**Definition C.11** (Stopping Time). *As we introduce in Remark C.10, we define the stopping time as $T_0 = N^{-R_0}$, where $R_0$ is a parameter to be specified later, and consider solving the ODE backward in time from $t = 1$ down to $t = T_0$.*

**Definition C.12** (Reduced Cube). *Let $I^d = [-1, 1]^d$ denote the $d$-dimensional cube. To mitigate boundary effects when $N$ is large, we define the reduced cube as*

$$I_N^d := [-1 + N^{-(1-\kappa\delta)}, 1 - N^{-(1-\kappa\delta)}]^d,$$

*where the parameter $\kappa > 0$ will be specified later in Assumption C.15.*

**Assumption C.13** (Smoothness and support of $p_0$). *The target probability $P_0$ has support contained in $I^d$, and its probability density function $p_0$ satisfies*

$$p_0 \in B^s_{p',q'}(I^d) \quad \text{and} \quad p_0 \in B^{\widetilde{s}}_{p',q'}(I^d \setminus I^d_N) \quad \text{with} \quad \widetilde{s} \geq \max\{6s - 1, 1\}.$$

**Assumption C.14** (Boundedness away from 0 and above). *There exists a constant $C_0 > 0$ such that*

$$C_0^{-1} \leq p_0(x) \leq C_0 \quad \text{forall} \quad x \in I^d.$$

**Assumption C.15** (Form of $(\alpha_t, \beta_t)$ and their bounds). *There are constants $\kappa \geq \frac{1}{2}$, $b_0 > 0$, $\widetilde{\kappa} > 0$, and $\widetilde{b}_0 > 0$ such that, for sufficiently small $t \geq T_0$,*

$$\alpha_t = b_0, t^\kappa, \quad \text{and} \quad 1 - \beta_t = \widetilde{b}_0, t^{\widetilde{\kappa}}.$$

*Moreover, there exist $D_0 > 0$ and $K_0 > 0$ such that $\forall t \in [T_0, 1]$, we have*

$$D_0^{-1} \leq \alpha_t^2 + \beta_t^2 \leq D_0, \quad |\dot{\alpha}_t| + |\dot{\beta}_t| \leq N^{K_0}.$$

**Assumption C.16** (Additional bound in the critical case $\kappa = \frac{1}{2}$). *If $\kappa = \frac{1}{2}$, then there exist $b_1 > 0$ and $D_1 > 0$ such that, for all $0 \leq \gamma < R_0$,*

$$\int_{T_0}^{N^{-\gamma}} \{(\dot{\alpha}_t)^2 + (\dot{\beta}_t)^2\}\mathrm{d}t \leq D_1(\log N)^{b_1}.$$

**Assumption C.17** (Lipschitz bound on the first moment). *There is a constant $C_L > 0$ such that, for all $t \in [T_0, 1]$,*

$$\|\frac{\partial}{\partial x} \int y p_t(y|x)\mathrm{d}y\|_{\mathrm{op}} \leq C_L.$$

## C.5 APPROXIMATION ERROR FOR SMALL $t$

**Lemma C.18** (Theorem 7 in (Fukumizu et al., 2024)). *Under Assumptions C.13 C.14 C.15 C.16 and C.17, and if the following holds*

- $L = O(\log^4 N)$.

- $\|W\|_\infty = O(N \log^6 N)$

- $S = O(N \log^8 N)$

- $B = \exp(O(\log N \log\log N))$.

*Then there exists a neural network $\phi \in \mathcal{M}(L, W, S, B)$ such that, for sufficiently large $N$, we have*

$$\int \|\phi(x,t) - \dot{x}_t^{\mathrm{true}}\|_2^2 p_t(x)\mathrm{d}x \lesssim (\dot{\alpha}_t^2 \log N + \dot{\beta}_t^2)N^{-\frac{2s}{d}},$$

*holds for any $t \in [T_0, 3T_*]$. In addition, $\phi$ can be taken so we have*

$$\|\phi(\cdot,t)\|_\infty = O(|\dot{\alpha}_t|\sqrt{\log n} + |\dot{\beta}_t|).$$

## C.6 APPROXIMATION ERROR FOR LARGE $t$

**Lemma C.19** (Theorem 7 in (Fukumizu et al., 2024)). *Fix $t_* \in [T_*, 1]$ and let $\eta > 0$ be arbitrary, under Assumptions C.13 C.14 C.15 C.16 and C.17, and if the following holds*

- $L = O(\log^4 N)$.

- $\|W\|_\infty = O(N)$

- $S = O(t_*^{-d\kappa} N^{\delta\kappa})$

- $B = \exp(O(\log N \log\log N))$.

Then there exist a neural network $\phi \in \mathcal{M}(L, W, S, B)$ such that

$$\int \|\phi(x, t) - \dot{x}_t^{\text{true}}\|^2 p_t(x) \mathrm{d}x \lesssim (\dot{\alpha}_t^2 \log N + \dot{\beta}_t^2) N^{-\eta}.$$

holds for any $t \in [2t_*, 1]$. In addition, $\phi$ can be taken so we have

$$\|\phi(\cdot, t)\|_\infty = O(|\dot{\alpha}_t| \log N + |\dot{\beta}_t|).$$

# D  THEORY OF HIGHER ORDER FLOW MATCHING

We use $\frac{\mathrm{d}^k}{\mathrm{d}t^k} x_t^{\text{true}}$ to denote the $k$-th order derivative of $x_t^{\text{true}}$ with respect to $t$. Note that $\dot{x}_t^{\text{true}} := \frac{\mathrm{d}}{\mathrm{d}t} x_t^{\text{true}}$, and $\ddot{x}_t^{\text{true}} := \frac{\mathrm{d}^2}{\mathrm{d}t^2} x_t^{\text{true}}$.

## D.1  APPROXIMATION ERROR OF SECOND ORDER FLOW MATCHING FOR SMALL $t$

**Theorem D.1** (Approximation error of second order flow matching for small $t$, formal version of Theorem 5.1). *Under Assumptions C.13 C.14 C.15 C.16 and C.17, and if the following holds*

- $L = O(\log^4 N)$.

- $\|W\|_\infty = O(N \log^6 N)$

- $S = O(N \log^8 N)$

- $B = \exp(O(\log N \log \log N))$.

*Then there exists neural networks $\phi_1, \phi_2 \in \mathcal{M}(L, W, S, B)$ such that, for sufficiently large $N$, we have*

$$\int (\|\phi_1(x, t) - \dot{x}_t^{\text{true}}\|_2^2 + \|\phi_2(x, t) - \ddot{x}_t^{\text{true}}\|_2^2) p_t(x) \mathrm{d}x$$

$$\lesssim (\dot{\alpha}_t^2 \log N + \dot{\beta}_t^2) N^{-\frac{2s}{d}} + \mathop{\mathbb{E}}_{x \sim P_t}[\|\dot{x}_t^{\text{true}} - \ddot{x}_t^{\text{true}}\|_2^2]$$

*holds for any $t \in [T_0, 3T_*]$. In addition, $\phi_1, \phi_2$ can be taken so we have*

$$\|\phi_1(\cdot, t)\|_\infty = O(|\dot{\alpha}_t| \sqrt{\log n} + |\dot{\beta}_t|) \quad \text{and} \quad \|\phi_2(\cdot, t)\|_\infty = O(|\dot{\alpha}_t| \sqrt{\log n} + |\dot{\beta}_t|).$$

*Proof.* Suppose that $t \in [T_0, 3T_*]$. By Lemma C.18, there is $\phi_1 \in \mathcal{M}(L, W, S, B)$ such that

$$\int (\|\phi_1(x, t) - \dot{x}_t^{\text{true}}\|_2^2 p_t(x) \lesssim (\dot{\alpha}_t^2 \log N + \dot{\beta}_t^2) N^{-\frac{2s}{d}}. \tag{2}$$

Next, we can show that there exists some $\phi_2 \in \mathcal{M}(L, W, S, B)$ such that

$$\int \|\phi_2(x, t) - \ddot{x}_t^{\text{true}}\|_2^2 p_t(x) \mathrm{d}x = \int \|\phi_2(x, t) - \dot{x}_t^{\text{true}} + \dot{x}_t^{\text{true}} - \ddot{x}_t^{\text{true}}\|_2^2 p_t(x) \mathrm{d}x$$

$$\leq \int (\|\phi_2(x, t) - \dot{x}_t^{\text{true}}\|_2 + \|\dot{x}_t^{\text{true}} - \ddot{x}_t^{\text{true}}\|_2)^2 p_t(x) \mathrm{d}x$$

$$\leq \int 2(\|\phi_2(x, t) - \dot{x}_t^{\text{true}}\|_2^2 + \|\dot{x}_t^{\text{true}} - \ddot{x}_t^{\text{true}}\|_2^2) p_t(x) \mathrm{d}x$$

$$= 2 \int \|\phi_2(x, t) - \dot{x}_t^{\text{true}}\|_2^2 p_t(x) \mathrm{d}x + 2 \int \|\dot{x}_t^{\text{true}} - \ddot{x}_t^{\text{true}}\|_2^2 p_t(x) \mathrm{d}x$$

$$= 2 \int \|\phi_2(x, t) - \dot{x}_t^{\text{true}}\|_2^2 p_t(x) \mathrm{d}x + 2 \mathop{\mathbb{E}}_{x \sim P_t}[\|\dot{x}_t^{\text{true}} - \ddot{x}_t^{\text{true}}\|_2^2$$

$$\lesssim (\dot{\alpha}_t^2 \log N + \dot{\beta}_t^2) N^{-\frac{2s}{d}} + \mathop{\mathbb{E}}_{x \sim P_t}[\|\dot{x}_t^{\text{true}} - \ddot{x}_t^{\text{true}}\|_2^2] \tag{3}$$

where the first step follows from the basic algebra, the second step follows from the triangle inequality, the third step follows from $(a+b)^2 \le 2a^2 + 2b^2$, the fourth step follows from basic algebra, the fifth step follows from the definition of expectation, and the last step follows from Lemma C.18.

Finally, by Eq. (2) and Eq. (3), for any $t \in [T_0, 3T_*]$, we have

$$\int (\|\phi_1(x,t) - \dot{x}_t^{\text{true}}\|_2^2 + \|\phi_2(x,t) - \ddot{x}_t^{\text{true}}\|_2^2) p_t(x) \mathrm{d}x$$

$$\lesssim (\dot{\alpha}_t^2 \log N + \dot{\beta}_t^2) N^{-\frac{2s}{d}} + \mathbb{E}_{x \sim P_t} [\|\dot{x}_t^{\text{true}} - \ddot{x}_t^{\text{true}}\|_2^2].$$

Moreover, by Lemma C.18, $\phi_1, \phi_2$ can be taken so we have

$$\|\phi_1(\cdot, t)\|_\infty = O(|\dot{\alpha}_t|\sqrt{\log n} + |\dot{\beta}_t|) \quad \text{and} \quad \|\phi_2(\cdot, t)\|_\infty = O(|\dot{\alpha}_t|\sqrt{\log n} + |\dot{\beta}_t|).$$

Thus, the proof is complete. $\square$

## D.2 APPROXIMATION ERROR OF HIGHER ORDER FLOW MATCHING FOR SMALL $t$

**Theorem D.2** (Approximation error of higher order flow matching for small $t$). *Under Assumptions C.13 C.14 C.15 C.16 and C.17, and if the following holds*

- $L = O(\log^4 N)$.

- $\|W\|_\infty = O(N \log^6 N)$

- $S = O(N \log^8 N)$

- $B = \exp(O(\log N \log \log N))$

- $K = O(1)$

*Then there exists neural networks $\phi_1, \phi_2, \dots, \phi_K \in \mathcal{M}(L, W, S, B)$ such that, for sufficiently large $N$, we have*

$$\int (\sum_{k=1}^K \|\phi_k(x,t) - \frac{\mathrm{d}^k}{\mathrm{d}t^k} x_t^{\text{true}}\|_2^2) p_t(x) \mathrm{d}x$$

$$\lesssim (\dot{\alpha}_t^2 \log N + \dot{\beta}_t^2) N^{-\frac{2s}{d}} + \sum_{k=1}^{K-1} \mathbb{E}_{x \sim P_t} [\|\frac{\mathrm{d}^k}{\mathrm{d}t^k} x_t^{\text{true}} - \frac{\mathrm{d}^{k+1}}{\mathrm{d}t^{k+1}} x_t^{\text{true}}\|_2^2]$$

*holds for any $t \in [T_0, 3T_*]$. In addition, for any $k \in [K]$, $\phi_k$ can be taken so we have*

$$\|\phi_k(\cdot, t)\|_\infty = O(|\dot{\alpha}_t|\sqrt{\log n} + |\dot{\beta}_t|).$$

*Proof.* We first show that for any $k \ge 2$, for any $t \in [T_0, 3T_*]$, there exists $\phi \in \mathcal{M}(L, W, S, B)$ such that

$$\int \|\phi(x,t) - \frac{\mathrm{d}^k}{\mathrm{d}t^k} x_t^{\text{true}}\|_2^2 p_t(x) \mathrm{d}x$$

$$\lesssim (\dot{\alpha}_t^2 \log N + \dot{\beta}_t^2) N^{-\frac{2s}{d}} + \sum_{j=1}^k \mathbb{E}_{x \sim P_t} [\|\frac{\mathrm{d}^j}{\mathrm{d}t^j} x_t^{\text{true}} - \frac{\mathrm{d}^{j+1}}{\mathrm{d}t^{j+1}} x_t^{\text{true}}\|_2^2]. \tag{4}$$

We prove this by mathematical induction.

**Base case.** The statements hold when $k = 2$ because of Lemma D.1.

**Induction step.** We assume that the statement hold for $k \ge 2$. We would like to show that it holds for $k+1$. We can show that, for any $t \in [T_0, 3T_*]$, there exists $\phi \in \mathcal{M}(L, S, W, B)$ such that

$$\int \|\phi(x,t) - \frac{\mathrm{d}^{k+1}}{\mathrm{d}t^{k+1}} x_t^{\text{true}}\|_2^2 p_t(x) \mathrm{d}x$$

$$= \int \|\phi(x,t) - \frac{\mathrm{d}^k}{\mathrm{d}t^k}x_t^{\mathrm{true}} + \frac{\mathrm{d}^k}{\mathrm{d}t^k}x_t^{\mathrm{true}} - \frac{\mathrm{d}^{k+1}}{\mathrm{d}t^{k+1}}x_t^{\mathrm{true}}\|_2^2 p_t(x)\mathrm{d}x$$

$$\leq \int (\|\phi(x,t) - \frac{\mathrm{d}^k}{\mathrm{d}t^k}x_t^{\mathrm{true}}\|_2 + \|\frac{\mathrm{d}^k}{\mathrm{d}t^k}x_t^{\mathrm{true}} - \frac{\mathrm{d}^{k+1}}{\mathrm{d}t^{k+1}}x_t^{\mathrm{true}}\|_2)^2 p_t(x)\mathrm{d}x$$

$$\leq \int 2(\|\phi(x,t) - \frac{\mathrm{d}^k}{\mathrm{d}t^k}x_t^{\mathrm{true}}\|_2^2 + \|\frac{\mathrm{d}^k}{\mathrm{d}t^k}x_t^{\mathrm{true}} - \frac{\mathrm{d}^{k+1}}{\mathrm{d}t^{k+1}}x_t^{\mathrm{true}}\|_2^2) p_t(x)\mathrm{d}x$$

$$= 2\int \|\phi(x,t) - \frac{\mathrm{d}^k}{\mathrm{d}t^k}x_t^{\mathrm{true}}\|_2^2 p_t(x)\mathrm{d}x + 2\int \|\frac{\mathrm{d}^k}{\mathrm{d}t^k}x_t^{\mathrm{true}} - \frac{\mathrm{d}^{k+1}}{\mathrm{d}t^{k+1}}x_t^{\mathrm{true}}\|_2^2 p_t(x)\mathrm{d}x$$

$$= 2\int \|\phi(x,t) - \frac{\mathrm{d}^k}{\mathrm{d}t^k}x_t^{\mathrm{true}}\|_2^2 p_t(x)\mathrm{d}x + 2\mathop{\mathbb{E}}_{x\sim P_t}[\|\frac{\mathrm{d}^k}{\mathrm{d}t^k}x_t^{\mathrm{true}} - \frac{\mathrm{d}^{k+1}}{\mathrm{d}t^{k+1}}x_t^{\mathrm{true}}\|_2^2]$$

$$\lesssim (\dot{\alpha}_t^2 \log N + \dot{\beta}_t^2)N^{-\frac{2s}{d}} + \sum_{j=1}^k \mathop{\mathbb{E}}_{x\sim P_t}[\|\frac{\mathrm{d}^j}{\mathrm{d}t^j}x_t^{\mathrm{true}} - \frac{\mathrm{d}^{j+1}}{\mathrm{d}t^{j+1}}x_t^{\mathrm{true}}\|_2^2] + \mathop{\mathbb{E}}_{x\sim P_t}[\|\frac{\mathrm{d}^k}{\mathrm{d}t^k}x_t^{\mathrm{true}} - \frac{\mathrm{d}^{k+1}}{\mathrm{d}t^{k+1}}x_t^{\mathrm{true}}\|_2^2]$$

$$= (\dot{\alpha}_t^2 \log N + \dot{\beta}_t^2)N^{-\frac{2s}{d}} + \sum_{j=1}^{k+1} \mathop{\mathbb{E}}_{x\sim P_t}[\|\frac{\mathrm{d}^j}{\mathrm{d}t^j}x_t^{\mathrm{true}} - \frac{\mathrm{d}^{j+1}}{\mathrm{d}t^{j+1}}x_t^{\mathrm{true}}\|_2^2], \tag{5}$$

where the first step follows from basic algebra, the second step follows from triangle inequality, the third step follows from the Cauchy-Schwarz inequality, the fourth step follows from basic algebra, the fifth step follows from the definition of expectation, the six step follows from Eq. (4).

Hence, there exists $\phi_1, \phi_2, \ldots, \phi_K \in \mathcal{M}(L, W, S, B)$ such that for $k \in [K]$, for any $t \in [T_0, 3T_*]$, we have

$$\int \|\phi_k(x,t) - \frac{\mathrm{d}^k}{\mathrm{d}t^k}x_t^{\mathrm{true}}\|_2^2 p_t(x)\mathrm{d}x$$

$$\lesssim (\dot{\alpha}_t^2 \log N + \dot{\beta}_t^2)N^{-\frac{2s}{d}} + \sum_{j=1}^k \mathop{\mathbb{E}}_{x\sim P_t}[\|\frac{\mathrm{d}^j}{\mathrm{d}t^j}x_t^{\mathrm{true}} - \frac{\mathrm{d}^{j+1}}{\mathrm{d}t^{j+1}}x_t^{\mathrm{true}}\|_2^2]. \tag{6}$$

Taking the summation over $k \in [K]$, we have for any $t \in [T_0, 3T_*]$,

$$\int \sum_{k=1}^K \|\phi_k(x,t) - \frac{\mathrm{d}^k}{\mathrm{d}t^k}x_t^{\mathrm{true}}\|_2^2 p_t(x)\mathrm{d}x$$

$$\lesssim K \cdot (\dot{\alpha}_t^2 \log N + \dot{\beta}_t^2)N^{-\frac{2s}{d}} + \sum_{k=1}^K (k \cdot \mathop{\mathbb{E}}_{x\sim P_t}[\|\frac{\mathrm{d}^j}{\mathrm{d}t^j}x_t^{\mathrm{true}} - \frac{\mathrm{d}^{j+1}}{\mathrm{d}t^{j+1}}x_t^{\mathrm{true}}\|_2^2])$$

$$\lesssim ((\dot{\alpha}_t)^2 \log N + (\dot{\beta}_t)^2)N^{-\frac{2s}{d}} + \sum_{k=1}^K \mathop{\mathbb{E}}_{x\sim P_t}[\|\frac{\mathrm{d}^j}{\mathrm{d}t^j}x_t^{\mathrm{true}} - \frac{\mathrm{d}^{j+1}}{\mathrm{d}t^{j+1}}x_t^{\mathrm{true}}\|_2^2]$$

where the first step follows from Eq. (6), and the second step uses $K = O(1)$.

Moreover, by Lemma C.19, $\phi_1, \phi_2, \ldots, \phi_K$ can be taken so we have for $k \in [K]$,

$$\|\phi_k(\cdot, t)\|_\infty = O(|\dot{\alpha}_t| \log \sqrt{n} + |\dot{\beta}_t|).$$

Thus, the proof is complete. $\qquad\square$

### D.3 APPROXIMATION ERROR OF SECOND ORDER FLOW MATCHING FOR LARGE $t$

**Theorem D.3** (Approximation error of second order flow matching for large $t$, formal version of Theorem 5.2). *Fix $t_* \in [T_*, 1]$ and let $\eta > 0$ be arbitrary, under Assumptions C.13 C.14 C.15 C.16 and C.17, and if the following holds*

- $L = O(\log^4 N)$.

- $\|W\|_\infty = O(N)$

- $S = O(t_*^{-d\kappa} N^{\delta\kappa})$

- $B = \exp(O(\log N \log \log N))$.

*Then there exist neural networks $\phi_1, \phi_2 \in \mathcal{M}(L, W, S, B)$ such that*

$$\int (\|\phi_1(x, t) - \dot{x}_t^{\text{true}}\|_2^2 + \|\phi_2(x, t) - \ddot{x}_t^{\text{true}}\|_2^2) p_t(x) \mathrm{d}x$$
$$\lesssim (\dot{\alpha}_t^2 \log N + \dot{\beta}_t^2) N^{-\eta} + \mathbb{E}_{x \sim P_t}[\|\dot{x}_t^{\text{true}} - \ddot{x}_t^{\text{true}}\|_2^2]$$

*holds for any $t \in [2t_*, 1]$. In addition, $\phi_1, \phi_2$ can be taken so we have*

$$\|\phi_1(\cdot, t)\|_\infty = O(|\dot{\alpha}_t| \log N + |\dot{\beta}_t|) \quad \text{and} \quad \|\phi_2(\cdot, t)\|_\infty = O(|\dot{\alpha}_t| \log N + |\dot{\beta}_t|).$$

*Proof.* Suppose that $t \in [2t_*, 1]$. By Lemma C.19, there is $\phi_1 \in \mathcal{M}(L, W, S, B)$ such that

$$\int (\|\phi_1(x, t) - \dot{x}_t^{\text{true}}\|_2^2 p_t(x) \mathrm{d}x \lesssim (\dot{\alpha}_t^2 \log N + \dot{\beta}_t^2) N^{-\eta}. \tag{7}$$

Next, we can show that there exists some $\phi_2 \in \mathcal{M}(L, W, S, B)$ such that

$$\int \|\phi_2(x, t) - \ddot{x}_t^{\text{true}}\|_2^2 p_t(x) \mathrm{d}x = \int \|\phi_2(x, t) - \dot{x}_t^{\text{true}} + \dot{x}_t^{\text{true}} - \ddot{x}_t^{\text{true}}\|_2^2 p_t(x) \mathrm{d}x$$

$$\leq \int (\|\phi_2(x, t) - \dot{x}_t^{\text{true}}\|_2 + \|\dot{x}_t^{\text{true}} - \ddot{x}_t^{\text{true}}\|_2)^2 p_t(x) \mathrm{d}x$$

$$\leq \int 2(\|\phi_2(x, t) - \dot{x}_t^{\text{true}}\|_2^2 + \|\dot{x}_t^{\text{true}} - \ddot{x}_t^{\text{true}}\|_2^2) p_t(x) \mathrm{d}x$$

$$= 2 \int \|\phi_2(x, t) - \dot{x}_t^{\text{true}}\|_2^2 p_t(x) \mathrm{d}x + 2 \int \|_2 \dot{x}_t^{\text{true}} - \ddot{x}_t^{\text{true}}\|_2^2 p_t(x) \mathrm{d}x$$

$$= 2 \int \|\phi_2(x, t) - \dot{x}_t^{\text{true}}\|_2^2 p_t(x) \mathrm{d}x + 2 \mathbb{E}_{x \sim P_t}[\|\dot{x}_t^{\text{true}} - \ddot{x}_t^{\text{true}}\|_2^2$$

$$\lesssim (\dot{\alpha}_t^2 \log N + \dot{\beta}_t^2) N^{-\eta} + \mathbb{E}_{x \sim P_t}[\|\dot{x}_t^{\text{true}} - \ddot{x}_t^{\text{true}}\|_2^2] \tag{8}$$

where the first step follows from the basic algebra, the second step follows from the triangle inequality, the third step follows from $(a + b)^2 \leq 2a^2 + 2b^2$, the fourth step follows from basic algebra, the fifth step follows from the definition of expectation, and the last step follows from Lemma C.19.

Finally, by Eq. (7) and Eq. (8), we have

$$\int (\|\phi_1(x, t) - \dot{x}_t^{\text{true}}\|^2 + \|\phi_2(x, t) - \ddot{x}_t^{\text{true}}\|^2) p_t(x) \mathrm{d}x$$
$$\lesssim (\dot{\alpha}_t^2 \log N + \dot{\beta}_t^2) N^{-\eta} + \mathbb{E}_{x \sim P_t}[\|\dot{x}_t^{\text{true}} - \ddot{x}_t^{\text{true}}\|^2].$$

Moreover, by Lemma C.19, $\phi_1, \phi_2$ can be taken so we have

$$\|\phi_1(\cdot, t)\|_\infty = O(|\dot{\alpha}_t| \log N + |\dot{\beta}_t|) \quad \text{and} \quad \|\phi_2(\cdot, t)\|_\infty = O(|\dot{\alpha}_t| \log N + |\dot{\beta}_t|).$$

Thus, the proof is complete. $\square$

### D.4 Approximation Error of Higher Order Flow Matching for Large $t$

**Theorem D.4** (Approximation error of higher order flow matching for large $t$). *Fix $t_* \in [T_*, 1]$ and let $\eta > 0$ be arbitrary, under Assumptions C.13 C.14 C.15 C.16 and C.17, and if the following holds*

- $L = O(\log^4 N)$.

- $\|W\|_\infty = O(N)$

- $S = O(t_*^{-d\kappa} N^{\delta\kappa})$

- $B = \exp(O(\log N \log\log N))$

- $K = O(1)$

*Then there exist neural networks $\phi_1, \phi_2, \ldots, \phi_K \in \mathcal{M}(L, W, S, B)$ such that,*

$$\int \left(\sum_{k=1}^{K} \|\phi_k(x,t) - \frac{\mathrm{d}^k}{\mathrm{d}t^k} x_t^{\mathrm{true}}\|^2\right) p_t(x)\mathrm{d}x$$

$$\lesssim (\dot{\alpha}_t^2 \log N + \dot{\beta}_t^2) N^{-\eta} + \sum_{k=1}^{K-1} \mathbb{E}_{x\sim P_t}[\|\frac{\mathrm{d}^k}{\mathrm{d}t^k} x_t^{\mathrm{true}} - \frac{\mathrm{d}^{k+1}}{\mathrm{d}t^{k+1}} x_t^{\mathrm{true}}\|^2]$$

*holds for any $t \in [2t_*, 1]$. In addition, for any $k \in [K]$, $\phi_k$ can be taken so we have*

$$\|\phi_k(\cdot, t)\|_\infty = O(|\dot{\alpha}_t| \log N + |\dot{\beta}_t|).$$

*Proof.* We first show that for any $k \geq 2$, for any $t \in [2t_*, 1]$, there exists $\phi \in \mathcal{M}(L, W, S, B)$ such that

$$\int \|\phi(x,t) - \frac{\mathrm{d}^k}{\mathrm{d}t^k} x_t^{\mathrm{true}}\|_2^2 p_t(x)\mathrm{d}x$$

$$\lesssim (\dot{\alpha}_t^2 \log N + \dot{\beta}_t^2) N^{-\eta} + \sum_{j=1}^{k} \mathbb{E}_{x\sim P_t}[\|\frac{\mathrm{d}^j}{\mathrm{d}t^j} x_t^{\mathrm{true}} - \frac{\mathrm{d}^{j+1}}{\mathrm{d}t^{j+1}} x_t^{\mathrm{true}}\|^2]. \qquad (9)$$

We prove this by mathematical induction.

**Base case.** The statements hold when $k = 2$ because of Lemma D.3.

**Induction step.** We assume that the statement hold for $k \geq 2$. We would like to show that it holds for $k + 1$. We can show that, for any $t \in [2t_*, 1]$, there exists $\phi \in \mathcal{M}(L, S, W, B)$ such that

$$\int \|\phi(x,t) - \frac{\mathrm{d}^{k+1}}{\mathrm{d}t^{k+1}} x_t^{\mathrm{true}}\|_2^2 p_t(x)\mathrm{d}x$$

$$= \int \|\phi(x,t) - \frac{\mathrm{d}^k}{\mathrm{d}t^k} x_t^{\mathrm{true}} + \frac{\mathrm{d}^k}{\mathrm{d}t^k} x_t^{\mathrm{true}} - \frac{\mathrm{d}^{k+1}}{\mathrm{d}t^{k+1}} x_t^{\mathrm{true}}\|_2^2 p_t(x)\mathrm{d}x$$

$$\leq \int (\|\phi(x,t) - \frac{\mathrm{d}^k}{\mathrm{d}t^k} x_t^{\mathrm{true}}\|_2 + \|\frac{\mathrm{d}^k}{\mathrm{d}t^k} x_t^{\mathrm{true}} - \frac{\mathrm{d}^{k+1}}{\mathrm{d}t^{k+1}} x_t^{\mathrm{true}}\|_2)^2 p_t(x)\mathrm{d}x$$

$$\leq \int 2(\|\phi(x,t) - \frac{\mathrm{d}^k}{\mathrm{d}t^k} x_t^{\mathrm{true}}\|_2^2 + \|\frac{\mathrm{d}^k}{\mathrm{d}t^k} x_t^{\mathrm{true}} - \frac{\mathrm{d}^{k+1}}{\mathrm{d}t^{k+1}} x_t^{\mathrm{true}}\|_2^2) p_t(x)\mathrm{d}x$$

$$= 2\int \|\phi(x,t) - \frac{\mathrm{d}^k}{\mathrm{d}t^k} x_t^{\mathrm{true}}\|_2^2 p_t(x)\mathrm{d}x + 2\int \|\frac{\mathrm{d}^k}{\mathrm{d}t^k} x_t^{\mathrm{true}} - \frac{\mathrm{d}^{k+1}}{\mathrm{d}t^{k+1}} x_t^{\mathrm{true}}\|_2^2 p_t(x)\mathrm{d}x$$

$$= 2\int \|\phi(x,t) - \frac{\mathrm{d}^k}{\mathrm{d}t^k} x_t^{\mathrm{true}}\|_2^2 p_t(x)\mathrm{d}x + 2\mathbb{E}_{x\sim P_t}[\|\frac{\mathrm{d}^k}{\mathrm{d}t^k} x_t^{\mathrm{true}} - \frac{\mathrm{d}^{k+1}}{\mathrm{d}t^{k+1}} x_t^{\mathrm{true}}\|_2^2]$$

$$\lesssim (\dot{\alpha}_t^2 \log N + \dot{\beta}_t^2) N^{-\eta} + \sum_{j=1}^{k} \mathbb{E}_{x\sim P_t}[\|\frac{\mathrm{d}^j}{\mathrm{d}t^j} x_t^{\mathrm{true}} - \frac{\mathrm{d}^{j+1}}{\mathrm{d}t^{j+1}} x_t^{\mathrm{true}}\|_2^2] + \mathbb{E}_{x\sim P_t}[\|\frac{\mathrm{d}^k}{\mathrm{d}t^k} x_t^{\mathrm{true}} - \frac{\mathrm{d}^{k+1}}{\mathrm{d}t^{k+1}} x_t^{\mathrm{true}}\|_2^2]$$

$$= (\dot{\alpha}_t^2 \log N + \dot{\beta}_t^2) N^{-\eta} + \sum_{j=1}^{k+1} \mathbb{E}_{x\sim P_t}[\|\frac{\mathrm{d}^j}{\mathrm{d}t^j} x_t^{\mathrm{true}} - \frac{\mathrm{d}^{j+1}}{\mathrm{d}t^{j+1}} x_t^{\mathrm{true}}\|_2^2], \qquad (10)$$

where the first step follows from basic algebra, the second step follows from triangle inequality, the third step follows from the Cauchy-Schwarz inequality, the fourth step follows from basic algebra, the fifth step follows from the definition of expectation, the six step follows from Eq. (9).

Hence, there exists $\phi_1, \phi_2, \ldots, \phi_K \in \mathcal{M}(L, W, S, B)$ such that for $k \in [K]$, for any $t \in [2t_*, 1]$, we have

$$\int \|\phi_k(x,t) - \frac{\mathrm{d}^k}{\mathrm{d}t^k} x_t^{\mathrm{true}}\|_2^2 p_t(x)\mathrm{d}x$$

$$\lesssim (\dot{\alpha}_t^2 \log N + \dot{\beta}_t^2) N^{-\eta} + \sum_{j=1}^{k} \mathbb{E}_{x \sim P_t} [\|\frac{\mathrm{d}^j}{\mathrm{d}t^j} x_t^{\text{true}} - \frac{\mathrm{d}^{j+1}}{\mathrm{d}t^{j+1}} x_t^{\text{true}}\|_2^2]. \tag{11}$$

Taking the summation over $k \in [K]$, we have for any $t \in [2t_*, 1]$,

$$\int \sum_{k=1}^{K} \|\phi_k(x, t) - \frac{\mathrm{d}^k}{\mathrm{d}t^k} x_t^{\text{true}}\|_2^2 p_t(x) \mathrm{d}x$$

$$\lesssim ((\dot{\alpha}_t)^2 \log N + (\dot{\beta}_t)^2) N^{-\eta} + \sum_{k=1}^{K} (k \cdot \mathbb{E}_{x \sim P_t}[\|\frac{\mathrm{d}^j}{\mathrm{d}t^j} x_t^{\text{true}} - \frac{\mathrm{d}^{j+1}}{\mathrm{d}t^{j+1}} x_t^{\text{true}}\|_2^2])$$

$$\lesssim (\dot{\alpha}_t^2 \log N + \dot{\beta}_t^2) N^{-\eta} + \sum_{k=1}^{K} \mathbb{E}_{x \sim P_t}[\|\frac{\mathrm{d}^j}{\mathrm{d}t^j} x_t^{\text{true}} - \frac{\mathrm{d}^{j+1}}{\mathrm{d}t^{j+1}} x_t^{\text{true}}\|_2^2]$$

where the first step follows from Eq. (11), and the second step uses $K = O(1)$. Moreover, by Lemma C.19, $\phi_1, \phi_2, \ldots, \phi_K$ can be taken so we have for $k \in [K]$,

$$\|\phi_k(\cdot, t)\|_\infty = O(|\dot{\alpha}_t| \log N + |\dot{\beta}_t|).$$

Thus, the proof is complete. $\qquad\square$

# E    EMPIRICAL ABLATION STUDY

In Section E.1, we introduce the three Gaussian mixture distribution datasets—four-mode, five-mode, and eight-mode—used in our empirical ablation study, along with their configurations for source and target modes. The subsequent subsections analyze the impact of different optimization terms. Section E.2 evaluates the performance of HOMO optimized solely with the first-order term. Section E.3 examines the effect of using only the second-order term. Section E.4 assesses results when optimization is guided by the self-consistency term. Section E.5 explores the combined effect of first- and second-order terms, while Section E.6 investigates the combination of second-order and self-consistency terms. Through these analyses, we aim to dissect the contributions of individual and combined loss terms in achieving effective transport trajectories.

## E.1    DATASET

Here we introduce three datasets we use: four-mode, five-mode, and eight-mode Gaussian mixture distribution datasets; each Gaussian component has a variance of $0.3$. In the four-mode Gaussian mixture distribution, four source mode(**brown**) positioned at a distance $D_0 = 5$ from the origin, and four target mode(**indigo**) positioned at a distance $D_0 = 14$ from the origin, each mode sample 200 points. In five-mode Gaussian mixture distribution, five source mode(**brown**) positioned at a distance $D_0 = 6$ from the origin, and five target mode(**indigo**) positioned at a distance $D_0 = 13$ from the origin, each mode sample 200 points. And in eight-mode Gaussian mixture distribution, eight source mode(**brown**) positioned at a distance $D_0 = 6$ from the origin, and eight target mode(**indigo**) positioned at a distance $D_0 = 13$ from the origin, each mode sample 100 points.

## E.2    ONLY FIRST ORDER TERM

We optimize models by the sum of squared error(SSE). The source distribution and target distribution are all Gaussian distributions. For the target transport trajectory setting, we follow the VP ODE framework from (Liu et al., 2022b), which is $x_t = \alpha_t x_0 + \beta_t x_1$. We choose $\alpha_t = \exp(-\frac{1}{4}a(1-t)^2 - \frac{1}{2}b(1-t))$ and $\beta_t = \sqrt{1 - \alpha_t^2}$, with hyperparameters $a = 19.9$ and $b = 0.1$. In the four-mode dataset, five-mode dataset, and eight-mode dataset, we all sample 100 points in each source mode and target mode. And in four-mode dataset training, we use an ODE solver and Adam optimizer, with 2 hidden layer MLP, 100 hidden dimensions, 800 batch size, 0.005 learning rate, and 1000 training steps. In five-mode dataset training, we also use an ODE solver and Adam optimizer, with 2 hidden layer MLP, 100 hidden dimensions, 1000 batch size, 0.005 learning rate, and 1000 training steps. And in eight-mode dataset training, we use an ODE solver and Adam optimizer, with 2 hidden layer MLP, 100 hidden dimensions, 1600 batch size, 0.005 learning rate, and 1000 training steps.

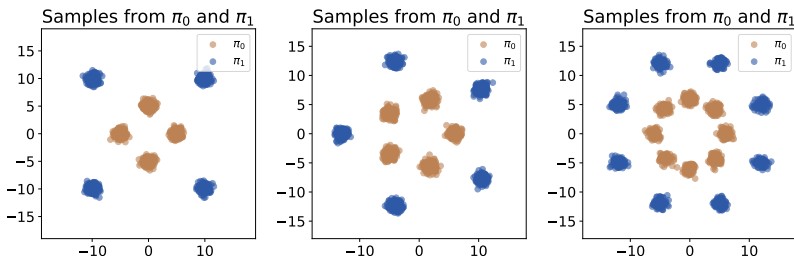

Figure 4: The four-mode Gaussian mixture distribution (**Left**), five-mode Gaussian mixture distribution (**Middle**), and eight-mode Gaussian mixture distribution (**Right**). Our goal is to make HOMO learn a transport trajectory from distribution $\pi_0$ (**brown**) to distribution $\pi_1$ (**indigo**).

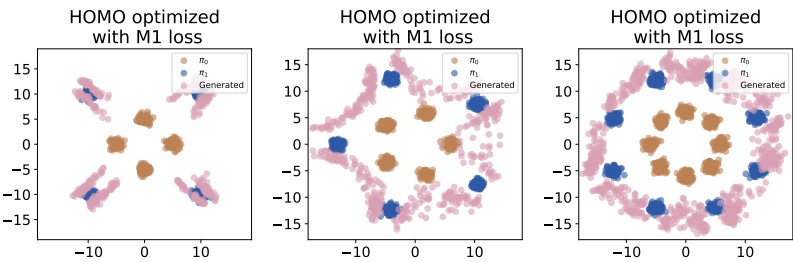

Figure 5: (A) The distributions generated by HOMO are only optimized by first-order term in four-mode dataset (**Left**), five-mode dataset (**Middle**), and eight-mode dataset (**Right**). The source distribution, $\pi_0$ (**brown**), and the target distribution, $\pi_1$ (**indigo**), are shown, along with the generated distribution (**pink**).

### E.3 ONLY SECOND ORDER TERM

We optimize models by the sum of squared error(SSE). The source distribution and target distribution are all Gaussian distributions. For the target transport trajectory setting, we follow the VP ODE framework from (Liu et al., 2022b), which is $x_t = \alpha_t x_0 + \beta_t x_1$. We choose $\alpha_t = \exp(-\frac{1}{4}a(1-t)^2 - \frac{1}{2}b(1-t))$ and $\beta_t = \sqrt{1 - \alpha_t^2}$, with hyperparameters $a = 19.9$ and $b = 0.1$. In the four-mode dataset, five-mode dataset, and eight-mode dataset, we all sample 100 points in each source mode and target mode. And in four-mode dataset training, we use an ODE solver and Adam optimizer, with 2 hidden layer MLP, 100 hidden dimensions, 800 batch size, 0.005 learning rate, and 100 training steps. In five-mode dataset training, we also use an ODE solver and Adam optimizer, with 2 hidden layer MLP, 100 hidden dimensions, 1000 batch size, 0.005 learning rate, and 100 training steps. And in eight-mode dataset training, we use an ODE solver and Adam optimizer, with 2 hidden layer MLP, 100 hidden dimensions, 1600 batch size, 0.005 learning rate, and 100 training steps.

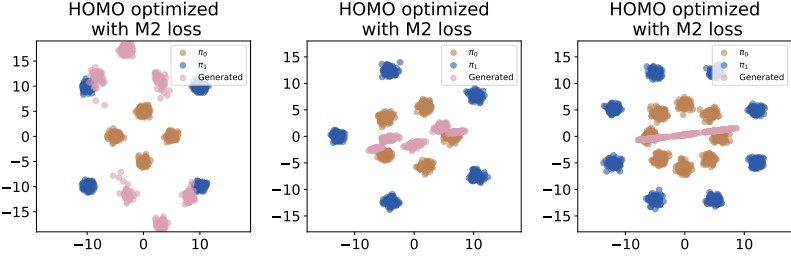

Figure 6: (B) The distributions generated by HOMO are only optimized by second-order term in the four-mode dataset (**Left**), five-mode dataset (**Middle**), and eight-mode dataset (**Right**). The source distribution, $\pi_0$ (**brown**), and the target distribution, $\pi_1$ (**indigo**), are shown, along with the generated distribution (**pink**).

### E.4 ONLY SELF-CONSISTENCY TERM

We optimize models by the sum of squared error(SSE). The source distribution and target distribution are all Gaussian distributions. For the target transport trajectory setting, we follow the VP ODE framework from (Liu et al., 2022b), which is $x_t = \alpha_t x_0 + \beta_t x_1$. We choose $\alpha_t = \exp(-\frac{1}{4}a(1-t)^2 - \frac{1}{2}b(1-t))$ and $\beta_t = \sqrt{1 - \alpha_t^2}$, with hyperparameters $a = 19.9$ and $b = 0.1$. In the four-mode dataset, five-mode dataset, and eight-mode dataset, we all sample 100 points in each source mode and target mode. And in four-mode dataset training, we use an ODE solver and Adam optimizer, with 2 hidden layer MLP, 100 hidden dimensions, 800 batch size, 0.005 learning rate, and 50 training steps. In five-mode dataset training, we also use an ODE solver and Adam optimizer, with 2 hidden layer MLP, 100 hidden dimensions, 1000 batch size, 0.005 learning rate, and 50 training steps. And in eight-mode dataset training, we use an ODE solver and Adam optimizer, with 2 hidden layer MLP, 100 hidden dimensions, 1600 batch size, 0.005 learning rate, and 50 training steps.

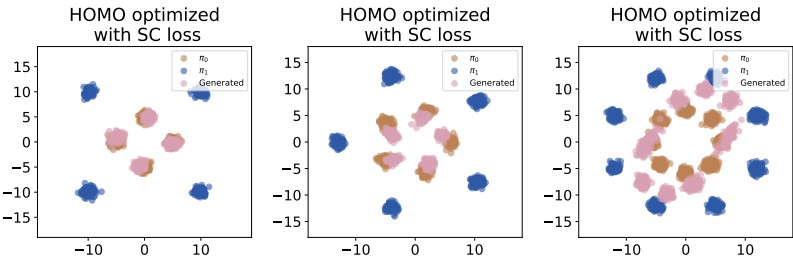

Figure 7: (C) The distributions generated by HOMO are only optimized by self-consistency term in the four-mode dataset (**Left**), five-mode dataset (**Middle**), and eight-mode dataset (**Right**). The source distribution, $\pi_0$ (**brown**), and the target distribution, $\pi_1$ (**indigo**), are shown, along with the generated distribution (**pink**).

### E.5 FIRST ORDER PLUS SECOND ORDER

We optimize models by the sum of squared error(SSE). The source distribution and target distribution are all Gaussian distributions. For the target transport trajectory setting, we follow the VP ODE framework from (Liu et al., 2022b), which is $x_t = \alpha_t x_0 + \beta_t x_1$. We choose $\alpha_t = \exp(-\frac{1}{4}a(1-t)^2 - \frac{1}{2}b(1-t))$ and $\beta_t = \sqrt{1 - \alpha_t^2}$, with hyperparameters $a = 19.9$ and $b = 0.1$. In the four-mode dataset, five-mode dataset, and eight-mode dataset, we all sample 100 points in each source mode and target mode. And in four-mode dataset training, we use an ODE solver and Adam optimizer, with 2 hidden layer MLP, 100 hidden dimensions, 800 batch size, 0.005 learning rate, and 1000 training steps. In five-mode dataset training, we also use an ODE solver and Adam optimizer, with 2 hidden layer MLP, 100 hidden dimensions, 1000 batch size, 0.005 learning rate, and 2000 training steps. And in eight-mode dataset training, we use an ODE solver and Adam optimizer, with 2 hidden layer MLP, 100 hidden dimensions, 1600 batch size, 0.005 learning rate, and 2000 training steps.

### E.6 SECOND ORDER PLUS SELF-TARGET

We optimize models by the sum of squared error(SSE). The source distribution and target distribution are all Gaussian distributions. For the target transport trajectory setting, we follow the VP ODE framework from (Liu et al., 2022b), which is $x_t = \alpha_t x_0 + \beta_t x_1$. We choose $\alpha_t = \exp(-\frac{1}{4}a(1-t)^2 - \frac{1}{2}b(1-t))$ and $\beta_t = \sqrt{1 - \alpha_t^2}$, with hyperparameters $a = 19.9$ and $b = 0.1$. In the four-mode dataset, five-mode dataset, and eight-mode dataset, we all sample 100 points in each source mode and target mode. And in four-mode dataset training, we use an ODE solver and Adam optimizer, with 2 hidden layer MLP, 100 hidden dimensions, 800 batch size, 0.005 learning rate, and 100 training steps. In five-mode dataset training, we also use an ODE solver and Adam optimizer, with 2 hidden layer MLP, 100 hidden dimensions, 1000 batch size, 0.005 learning rate, and 100 training steps. And in eight-mode dataset training, we use an ODE solver and Adam

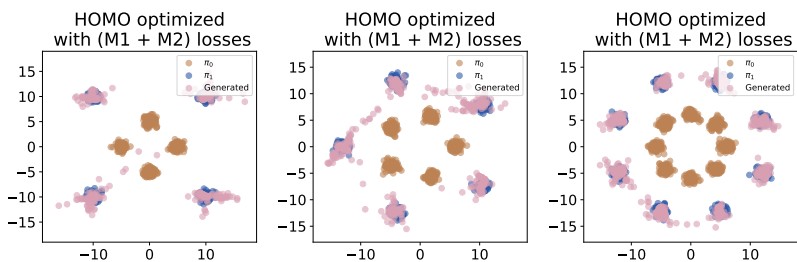

Figure 8: (A + B) The distributions generated by HOMO, optimized by first-order term and second-order term in four-mode dataset (**Left**), five-mode dataset (**Middle**), and eight-mode dataset (**Right**). The source distribution, $\pi_0$ (**brown**), and the target distribution, $\pi_1$ (**indigo**), are shown, along with the generated distribution (**pink**).

optimizer, with 2 hidden layer MLP, 100 hidden dimensions, 1600 batch size, 0.005 learning rate, and 100 training steps.

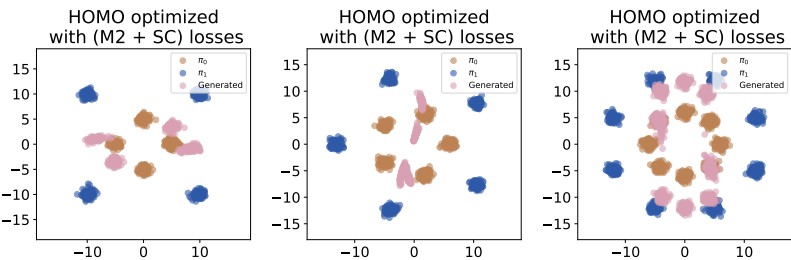

Figure 9: (B + C) The distributions generated by HOMO, optimized by second-order term and self-consistency term in four-mode dataset (**Left**), five-mode dataset (**Middle**), and eight-mode dataset (**Right**). The source distribution, $\pi_0$ (**brown**), and the target distribution, $\pi_1$ (**indigo**), are shown, along with the generated distribution (**pink**).

### E.7 FIRST ORDER PLUS SELF-TARGET

We optimize models by the sum of squared error(SSE). The source distribution and target distribution are all Gaussian distributions. For the target transport trajectory setting, we follow the VP ODE framework from (Liu et al., 2022b), which is $x_t = \alpha_t x_0 + \beta_t x_1$. We choose $\alpha_t = \exp(-\frac{1}{4}a(1-t)^2 - \frac{1}{2}b(1-t))$ and $\beta_t = \sqrt{1 - \alpha_t^2}$, with hyperparameters $a = 19.9$ and $b = 0.1$. In the four-mode dataset, five-mode dataset, and eight-mode dataset, we all sample 100 points in each source mode and target mode. And in four-mode dataset training, we use an ODE solver and Adam optimizer, with 2 hidden layer MLP, 100 hidden dimensions, 800 batch size, 0.005 learning rate, and 1000 training steps. In five-mode dataset training, we also use an ODE solver and Adam optimizer, with 2 hidden layer MLP, 100 hidden dimensions, 1000 batch size, 0.005 learning rate, and 1000 training steps. And in eight-mode dataset training, we use an ODE solver and Adam optimizer, with 2 hidden layer MLP, 100 hidden dimensions, 1600 batch size, 0.005 learning rate, and 1000 training steps.

### E.8 HOMO

We optimize models by the sum of squared error(SSE). The source distribution and target distribution are all Gaussian distributions. For the target transport trajectory setting, we follow the VP ODE framework from (Liu et al., 2022b), which is $x_t = \alpha_t x_0 + \beta_t x_1$. We choose $\alpha_t = \exp(-\frac{1}{4}a(1-t)^2 - \frac{1}{2}b(1-t))$ and $\beta_t = \sqrt{1 - \alpha_t^2}$, with hyperparameters $a = 19.9$ and $b = 0.1$. In the four-mode dataset, five-mode dataset, and eight-mode dataset, we all sample 100 points in each source mode and target mode. And in four-mode dataset training, we use an ODE solver and Adam optimizer, with 2 hidden layer MLP, 100 hidden dimensions, 800 batch size, 0.005

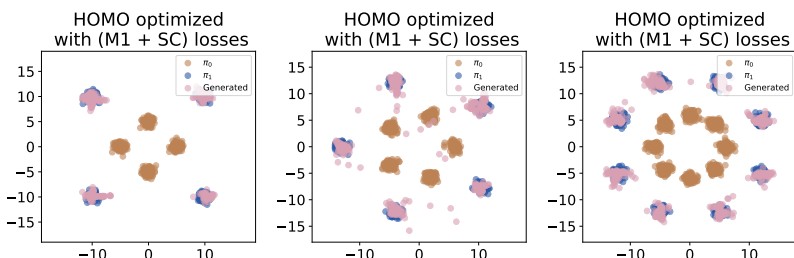

Figure 10: (A + C) The distributions generated by HOMO, optimized by first-order term and self-consistency term in four-mode dataset (**Left**), five-mode dataset (**Middle**), and eight-mode dataset (**Right**). The source distribution, $\pi_0$ (**brown**), and the target distribution, $\pi_1$ (**indigo**), are shown, along with the generated distribution (**pink**).

learning rate, and 1000 training steps. In five-mode dataset training, we also use an ODE solver and Adam optimizer, with 2 hidden layer MLP, 100 hidden dimensions, 1000 batch size, 0.005 learning rate, and 1000 training steps. And in eight-mode dataset training, we use an ODE solver and Adam optimizer, with 2 hidden layer MLP, 100 hidden dimensions, 1600 batch size, 0.005 learning rate, and 1000 training steps.

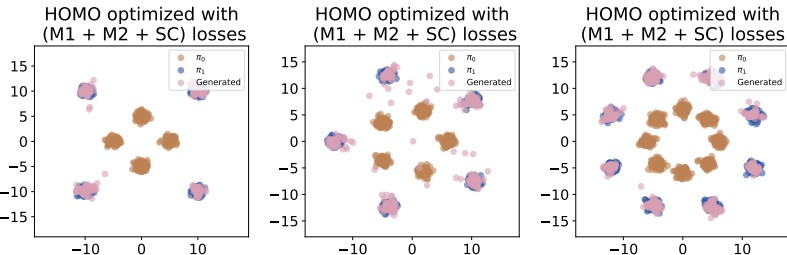

Figure 11: (A + B + C) The distributions generated by HOMO in four-mode dataset (**Left**), five-mode dataset (**Middle**), and eight-mode dataset (**Right**). The source distribution, $\pi_0$ (**brown**), and the target distribution, $\pi_1$ (**indigo**), are shown, along with the generated distribution (**pink**).

# F    COMPLEX DISTRIBUTION EXPERIMENT

In Section F.1, we introduce the datasets used in our experiments. The analysis of results with first-order and second-order terms in Section F.2, and we evaluate the performance with first-order and self-consistency terms in Section F.3, assess the impact of second-order and self-consistency terms in Section F.4. Finally, we present the overall results of HOMO with all loss terms combined in Section F.5.

## F.1    DATASETS

Here, we introduce four datasets we proposed: circle dataset, irregular ring dataset, spiral line dataset, and spin dataset. In the circle dataset, we sample 600 points from Gaussian distribution with 0.3 variance for both source distribution and target distribution. In the irregular ring dataset, we sample 600 points from Gaussian distribution with 0.3 variance for both source distribution and target distribution. In the spiral line dataset, we sample 600 points from Gaussian distribution with 0.3 variance for both source distribution and target distribution. In the spin dataset, we sample 600 points from the Gaussian distribution with 0.3 variance for both source distribution and target distribution.

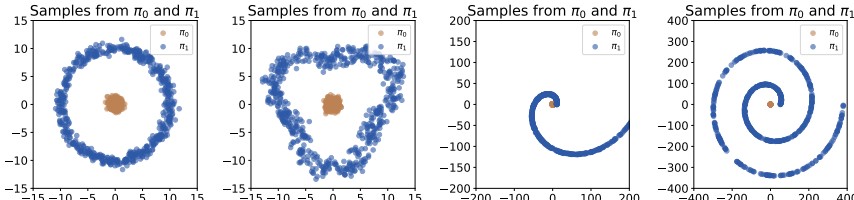

Figure 12: The circle dataset(**Left most**), irregular ring dataset (**Middle left**), spiral line dataset (**Middle right**), and spin dataset (**Right most**). Our goal is to make HOMO to learn a transport trajectory from distribution $\pi_0$ (**brown**) to distribution $\pi_1$ (**indigo**).

## F.2 FIRST ORDER PLUS SECOND ORDER

We optimize models by the sum of squared error(SSE). The source distribution and target distribution are all Gaussian distributions. For the target transport trajectory setting, we follow the VP ODE framework from (Liu et al., 2022b), which is $x_t = \alpha_t x_0 + \beta_t x_1$. We choose $\alpha_t = \exp(-\frac{1}{4}a(1-t)^2 - \frac{1}{2}b(1-t))$ and $\beta_t = \sqrt{1-\alpha_t^2}$, with hyperparameters $a = 19.9$ and $b = 0.1$. In the circle dataset, we all sample 400 points, both source distribution and target distribution. In the irregular ring dataset, we all sample 600 points, both source distribution and target distribution. In the spiral line dataset, we all sample 300 points, both source distribution and target distribution. In circle dataset training, we use an ODE solver and Adam optimizer, with 2 hidden layer MLP, 100 hidden dimensions, 800 batch size, 0.005 learning rate, and 1000 training steps. In irregular ring dataset training, we also use an ODE solver and Adam optimizer, with 2 hidden layer MLP, 100 hidden dimensions, 1000 batch size, 0.005 learning rate, and 1000 training steps. In spiral line dataset training, we use an ODE solver and Adam optimizer, with 2 hidden layer MLP, 100 hidden dimensions, 1600 batch size, 0.005 learning rate, and 1000 training steps. In spiral line dataset training, we use an ODE solver and Adam optimizer, with 2 hidden layer MLP, 100 hidden dimensions, 1600 batch size, 0.005 learning rate, and 1000 training steps.

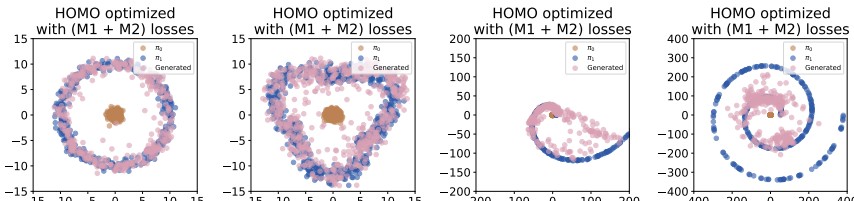

Figure 13: (M1+M2) **HOMO results on complex datasets with two kinds of loss: first-order and second-order terms.** The distributions generated by HOMO, in circle dataset(**Left most**), irregular ring dataset (**Middle left**), spiral line dataset (**Middle right**) and spin dataset (**Right most**). The source distribution, $\pi_0$ (**brown**), and the target distribution, $\pi_1$ (**indigo**), are shown, along with the generated distribution (**pink**).

## F.3 FIRST ORDER PLUS SELF-CONSISTENCY TERM

We optimize models by the sum of squared error(SSE). The source distribution and target distribution are all Gaussian distributions. For the target transport trajectory setting, we follow the VP ODE framework from (Liu et al., 2022b), which is $x_t = \alpha_t x_0 + \beta_t x_1$. We choose $\alpha_t = \exp(-\frac{1}{4}a(1-t)^2 - \frac{1}{2}b(1-t))$ and $\beta_t = \sqrt{1-\alpha_t^2}$, with hyperparameters $a = 19.9$ and $b = 0.1$. In the circle dataset, we all sample 400 points, both source distribution and target distribution. In the irregular ring dataset, we all sample 600 points, both source distribution and target distribution. In the spiral line dataset, we all sample 300 points, both source distribution and target distribution. In circle dataset training, we use an ODE solver and Adam optimizer, with 2 hidden layer MLP, 100 hidden dimensions, 800 batch size, 0.005 learning rate, and 1000 training steps. In irregular ring dataset training, we also use an ODE solver and Adam optimizer, with 2 hidden layer MLP, 100 hidden dimensions, 1000 batch size, 0.005 learning rate, and 1000 training steps. In

spiral line dataset training, we use an ODE solver and Adam optimizer, with 2 hidden layer MLP, 100 hidden dimensions, 1600 batch size, 0.005 learning rate, and 1000 training steps. In spiral line dataset training, we use an ODE solver and Adam optimizer, with 2 hidden layer MLP, 100 hidden dimensions, 1600 batch size, 0.005 learning rate, and 1000 training steps.

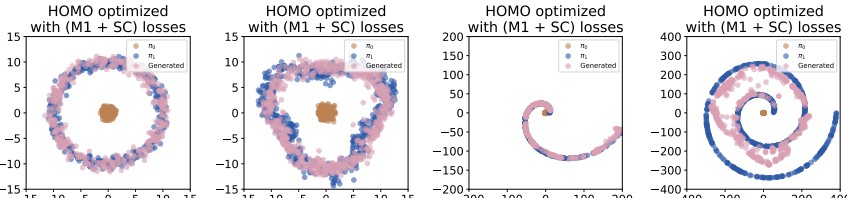

Figure 14: (M1+SC) **HOMO results on complex datasets with two kinds of loss: first-order and self-consistency terms.** The distributions generated by HOMO, in circle dataset(**Left most**), irregular ring dataset (**Middle left**), spiral line dataset (**Middle right**) and spin dataset (**Right most**). The source distribution, $\pi_0$ (**brown**), and the target distribution, $\pi_1$ (**indigo**), are shown, along with the generated distribution (**pink**).

### F.4 SECOND ORDER PLUS SELF-CONSISTENCY TERM

We optimize models by the sum of squared error(SSE). The source distribution and target distribution are all Gaussian distributions. For the target transport trajectory setting, we follow the VP ODE framework from (Liu et al., 2022b), which is $x_t = \alpha_t x_0 + \beta_t x_1$. We choose $\alpha_t = \exp(-\frac{1}{4}a(1-t)^2 - \frac{1}{2}b(1-t))$ and $\beta_t = \sqrt{1 - \alpha_t^2}$, with hyperparameters $a = 19.9$ and $b = 0.1$. In the circle dataset, we all sample 400 points, both source distribution and target distribution. In the irregular ring dataset, we all sample 600 points, both source distribution and target distribution. In the spiral line dataset, we all sample 300 points, both source distribution and target distribution. And in circle dataset training, we use an ODE solver and Adam optimizer, with 2 hidden layer MLP, 100 hidden dimensions, 800 batch size, 0.005 learning rate, and 100 training steps. In irregular ring dataset training, we also use an ODE solver and Adam optimizer, with 2 hidden layer MLP, 100 hidden dimensions, 100 batch size, 0.005 learning rate, and 1000 training steps. In spiral line dataset training, we use an ODE solver and Adam optimizer, with 2 hidden layer MLP, 100 hidden dimensions, 1600 batch size, 0.005 learning rate, and 100 training steps. In spiral line dataset training, we use an ODE solver and Adam optimizer, with 2 hidden layer MLP, 100 hidden dimensions, 1600 batch size, 0.005 learning rate, and 1000 training steps.

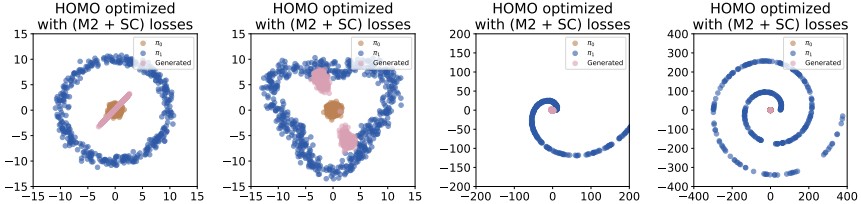

Figure 15: (M2+SC) **HOMO results on complex datasets with two kinds of loss: second-order and self-consistency terms.** The distributions generated by HOMO, in circle dataset(**Left most**), irregular ring dataset (**Middle left**), spiral line dataset (**Middle right**) and spin dataset (**Right most**). The source distribution, $\pi_0$ (**brown**), and the target distribution, $\pi_1$ (**indigo**), are shown, along with the generated distribution (**pink**).

### F.5 HOMO

We optimize models by the sum of squared error(SSE). The source distribution and target distribution are all Gaussian distributions. For the target transport trajectory setting, we follow the VP ODE framework from (Liu et al., 2022b), which is $x_t = \alpha_t x_0 + \beta_t x_1$. We choose

$\alpha_t = \exp(-\frac{1}{4}a(1-t)^2 - \frac{1}{2}b(1-t))$ and $\beta_t = \sqrt{1-\alpha_t^2}$, with hyperparameters $a = 19.9$ and $b = 0.1$. In the circle dataset, we all sample 400 points, both source distribution and target distribution. In the irregular ring dataset, we all sample 600 points, both source distribution and target distribution. In the spiral line dataset, we all sample 300 points, both source distribution and target distribution. And in circle dataset training, we use an ODE solver and Adam optimizer, with 2 hidden layer MLP, 100 hidden dimensions, 800 batch size, 0.005 learning rate, and 1000 training steps. In irregular ring dataset training, we also use an ODE solver and Adam optimizer, with 2 hidden layer MLP, 100 hidden dimensions, 1000 batch size, 0.005 learning rate, and 1000 training steps. In spiral line dataset training, we use an ODE solver and Adam optimizer, with 2 hidden layer MLP, 100 hidden dimensions, 1600 batch size, 0.005 learning rate, and 1000 training steps. In spiral line dataset training, we use an ODE solver and Adam optimizer, with 2 hidden layer MLP, 100 hidden dimensions, 1600 batch size, 0.005 learning rate, and 1000 training steps.

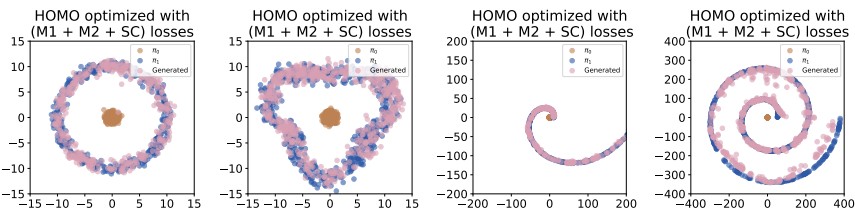

Figure 16: (M1+M2+SC) **HOMO results on complex datasets with three kinds of loss: first-order, second-order, and self-consistency terms.** The distributions generated by HOMO in circle dataset(**Left most**), irregular ring dataset (**Middle left**), spiral line dataset (**Middle right**), and spin dataset (**Right most**). The source distribution, $\pi_0$ (**brown**), and the target distribution, $\pi_1$ (**indigo**), are shown, along with the generated distribution (**pink**).

## G  THIRD-ORDER HOMO

This section extends HOMO to third-order dynamics and analyzes its performance on complex synthetic tasks. Section G.1 introduces the training and sampling algorithms incorporating third-order dynamics. Section G.2 compares two trajectory parameterization strategies for high-order systems. Section G.3 describes the 2 Round Spin, 3 Round Spin, and Dot-Circle datasets designed to test complex mode transitions. Section G.4 provides quantitative analysis through Euclidean distance metrics between generated and target distributions. Section G.5 evaluates the isolated impact of self-consistency constraints. Section G.6 examines first-order dynamics coupled with self-consistency regularization. Section G.7 studies the combined effect of first-, second-order dynamics and self-consistency. Finally, Section G.8 demonstrates full third-order HOMO with all optimization terms, analyzing trajectory linearity and mode fidelity under different trajectory settings.

### G.1  ALGORITHM

Here we first introduce the training algorithm of our third-order HOMO:

Then we will discuss the sampling algorithm in third-order HOMO:

### G.2  TRAJECTORY SETTING

We have trajectory as:

$$z_t = \alpha_t z_0 + \beta_t z_1$$

In original trajectory, we choose $\alpha_t = \exp(-\frac{1}{4}a(1-t)^2 - \frac{1}{2}b(1-t))$ and $\beta_t = \sqrt{1-\alpha_t^2}$, with hyperparameters $a = 19.9$ and $b = 0.1$. And new trajectory as $\alpha_t = 1 - (3t^2 - 2t^3)$ and $\beta_t = 3t^2 - 2t^3$.

---

**Algorithm 5** Third-Order HOMO Training

---

1: **while** not converged **do**
2:      $x_0 \sim \mathcal{N}(0, I), x_1 \sim D, (d, t) \sim p(d, t)$
3:      $\beta_t \leftarrow \sqrt{1 - \alpha_t^2}$
4:      $x_t \leftarrow \alpha_t \cdot x_0 + \beta_t \cdot x_1$                                ▷ Noise data point
5:      **for** first $k$ batch elements **do**
6:          $\dot{s}_t^{\text{true}} \leftarrow \dot{\alpha}_t x_0 + \dot{\beta}_t x_1$                          ▷ First-order target
7:          $\ddot{s}_t^{\text{true}} \leftarrow \ddot{\alpha}_t x_0 + \ddot{\beta}_t x_1$                        ▷ Second-order target
8:          $\dddot{s}_t^{\text{true}} \leftarrow \dddot{\alpha}_t x_0 + \dddot{\beta}_t x_1$                       ▷ Third-order target
9:          $d \leftarrow 0$
10:     **end for**
11:     **for** other batch elements **do**
12:          $s_t \leftarrow u_1(x_t, t, d)$                      ▷ First small step of first order
13:          $\dot{s}_t \leftarrow u_2(u_1(x_t, t, d), x_t, t, d)$          ▷ First small step of second order
14:          $\ddot{s}_t \leftarrow u_3(u_2(u_1(x_t, t, d), x_t, t, d), u_1(x_t, t, d), x_t, t, d)$    ▷ First small step of third order
15:          $x_{t+d} \leftarrow x_t + d \cdot s_t + \frac{d^2}{2} \dot{s}_t + \frac{d^6}{3} \ddot{s}_t$                    ▷ Follow ODE
16:          $s_{t+d} \leftarrow u_1(x_{t+d}, t + d, d)$               ▷ Second small step of first order
17:          $\dot{s}_t^{\text{target}} \leftarrow$ stopgrad $(s_t + s_{t+d})/2$      ▷ Self-consistency target of first order
18:     **end for**
19:     $\theta \leftarrow \nabla_\theta(\|u_1(x_t, t, 2d) - \dot{s}_t^{\text{true}}\|^2$
                 $+ \|u_2(u_1(x_t, t, 2d), x_t, t, 2d) - \ddot{s}_t^{\text{true}}\|^2$
                 $+ \|u_3(u_2(u_1(x, t, d), x, t, d), u_1(x, t, d), x, t, d) - \dddot{s}_t^{\text{true}}\|^2$
                 $+ \|u_1(x_t, t, 2d) - \dot{s}_t^{\text{target}}\|^2)$
20: **end while**

---

**Algorithm 6** Third-Order HOMO Sampling

---

1: $x \sim \mathcal{N}(0, I)$
2: $d \leftarrow 1/M$
3: $t \leftarrow 0$
4: **for** $n \in [0, \ldots, M - 1]$ **do**
5:     $x \leftarrow x + d \cdot u_1(x, t, d) + \frac{d^2}{2} \cdot u_2(u_1(x, t, d), x, t, d) + \frac{d^3}{6} \cdot u_3(u_2(u_1(x, t, d), x, t, d), u_1(x, t, d), x, t, d)$
6:     $t \leftarrow t + d$
7: **end for**
8: **return** $x$

---

### G.3 DATASET

Here, we introduce three datasets we use: 2 Round spin, 3 Round spin, and Dot-Circle datasets. In 2 Round spin dataset and 3 Round spin dataset, we both sample 600 points from Gaussian distribution with 0.3 variance for both source distribution and target distribution. In Dot-Circle datasets, we sample 300 points from the center dot and 300 points from the outermost circle, combine them as source distribution, and then sample 600 points from 2 round spin distribution.

### G.4 EUCLIDEAN DISTANCE LOSS

Here, we present the Euclidean distance loss performance of four different loss terms combined under the original trajectory setting and the new trajectory setting.

### G.5 ONLY SELF-CONSISTENCY TERM

We optimize models by the sum of squared error(SSE). The source distribution and target distribution are all Gaussian distributions. For the first line, we use the original transport trajectory setting, followed by the VP ODE framework from (Liu et al., 2022b), which is $x_t = \alpha_t x_0 + \beta_t x_1$. We choose $\alpha_t = \exp(-\frac{1}{4}a(1 - t)^2 - \frac{1}{2}b(1 - t))$ and $\beta_t = \sqrt{1 - \alpha_t^2}$, with hyperparameters $a = 19.9$

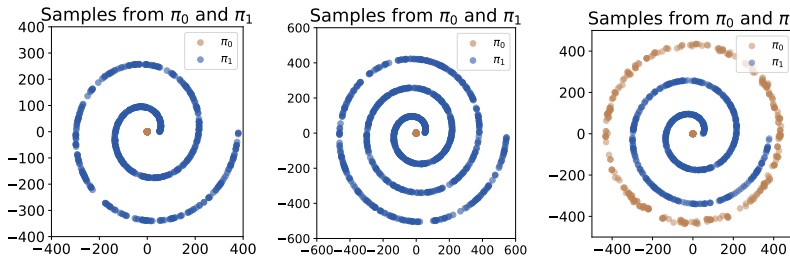

Figure 17: The 2 Round spin dataset(**Left**), 3 Round spin dataset(**Middle**), and Dot-Circle datasets(**Right**). Our goal is to make HOMO learn a transport trajectory from distribution $\pi_0$ (**brown**) to distribution $\pi_1$ (**indigo**).

Table 4: **Euclidean distance loss of three complex distribution datasets under new trajectory setting.** Lower values indicate more accurate distribution transfer results. Optimal values are highlighted in **Bold**. And Underlined numbers represent the second best (second lowest) loss value for each dataset (row). For the qualitative results of a mixture of Gaussian experiments, please refer to Figure 1.

| Loss terms | 2 Round spin | 3 Round spin | Dot-Circle |
|---|---|---|---|
| SC | 41.265 | 48.201 | 87.407 |
| M1 + SC | 14.926 | 18.376 | 30.027 |
| M1 + M2 + SC | 11.435 | 12.422 | 24.712 |
| M1 + M2 + SC + M3 | **4.701** | **9.261** | **21.968** |

and $b = 0.1$. In 2 Round spin datasets and 3 Round spin datasets, we sample 400 points, both source distribution and target distribution. In the Dot-Circle dataset, we sample 600 points from both source distribution and target distribution, 300 points of source points from the circle, and another 300 from the center dot. In 2-round dataset training, we use an ODE solver and Adam optimizer, with 2 hidden layer MLP, 100 hidden dimensions, 800 batch size, 0.005 learning rate, and 180 training steps. In 3-round spin dataset training, we also use an ODE solver and Adam optimizer, with 2 hidden layer MLP, 100 hidden dimensions, 1000 batch size, 0.005 learning rate, and 180 training steps. In Dot-Circle dataset training, we use an ODE solver and Adam optimizer, with 2 hidden layer MLP, 100 hidden dimensions, 1600 batch size, 0.005 learning rate, and 180 training steps.

### G.6 First Order Plus Self-Consistency

We optimize models by the sum of squared error(SSE). The source distribution and target distribution are all Gaussian distributions. For the first line, we use the original transport trajectory setting, followed by the VP ODE framework from (Liu et al., 2022b), which is $x_t = \alpha_t x_0 + \beta_t x_1$. We choose $\alpha_t = \exp(-\frac{1}{4}a(1-t)^2 - \frac{1}{2}b(1-t))$ and $\beta_t = \sqrt{1 - \alpha_t^2}$, with hyperparameters $a = 19.9$ and $b = 0.1$. In 2 Round spin datasets and 3 Round spin datasets, we sample 400 points in both source distribution and target distribution. And in the Dot-Circle dataset, we sample 600 points from both source distribution and target distribution, 300 points of sources points from the circle, and another 300 from the center dot. In 2 Round dataset training, we use ODE solver and Adam optimizer, with 2 hidden layer MLP, 100 hidden dimension, 800 batch size, 0.005 learning rate, and 1000 training steps. In 3 Round spin dataset training, we also use ODE solver and Adam optimizer, with 2 hidden layer MLP, 100 hidden dimension, 1000 batch size, 0.005 learning rate, and 2000 training steps. And in Dot-Circle dataset training, we use an ODE solver and Adam optimizer, with 2 hidden layer MLP, 100 hidden dimensions, 1600 batch size, 0.005 learning rate, and 10000 training steps.

### G.7 First Order Plus Second Order Plus Self-Consistency

We optimize models by the sum of squared error(SSE). The source distribution and target distribution are all Gaussian distributions. For the first line, we use the original transport trajectory setting,

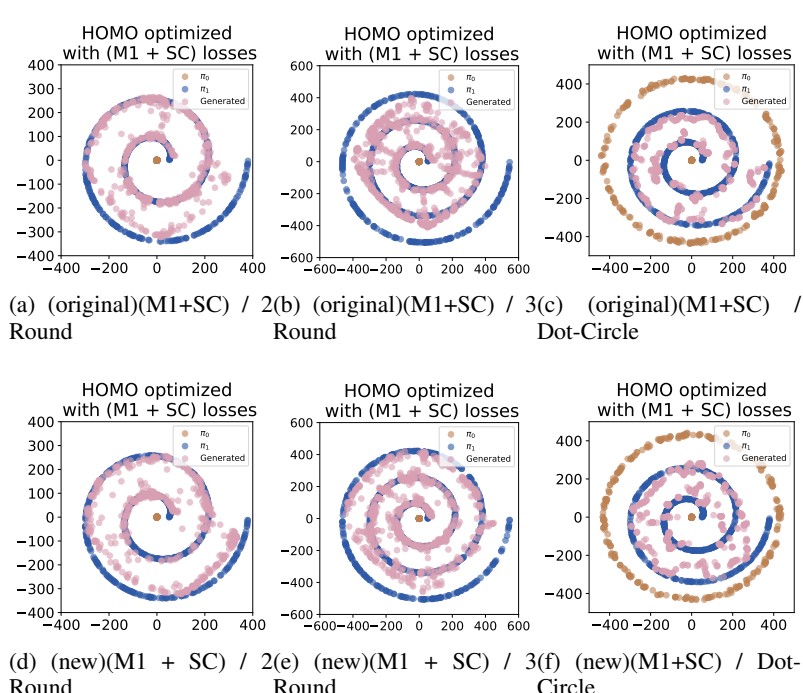

(a) (original)(M1+SC) / 2 Round    (b) (original)(M1+SC) / 3 Round    (c) (original)(M1+SC) / Dot-Circle

(d) (new)(M1 + SC) / 2 Round    (e) (new)(M1 + SC) / 3 Round    (f) (new)(M1+SC) / Dot-Circle

Figure 18: (M1+SC) The distributions generated by HOMO are only optimized by first-order loss and self-consistency loss. **Upper row(original trajectory setting):** Figure (a), in 2 Round spin dataset. Figure (b), in 3 Round spin dataset. Figure (c), in Dot-Circle dataset. **Lower row(new trajectory setting):** Figure (d), in 2 Round spin dataset. Figure (e), in 3 Round spin dataset. Figure (f), in Dot-Circle dataset. The source distribution, $\pi_0$ (**brown**), and the target distribution, $\pi_1$ (**indigo**), are shown, along with the generated distribution (**pink**).

followed by the VP ODE framework from (Liu et al., 2022b), which is $x_t = \alpha_t x_0 + \beta_t x_1$. We choose $\alpha_t = \exp(-\frac{1}{4}a(1-t)^2 - \frac{1}{2}b(1-t))$ and $\beta_t = \sqrt{1-\alpha_t^2}$, with hyperparameters $a = 19.9$ and $b = 0.1$. In 2 Round spin datasets and 3 Round spin datasets, we sample 400 points, both source distribution and target distribution. And in the Dot-Circle dataset, we sample 600 points from both source distribution and target distribution, 300 points of source points from the circle, and another 300 from the center dot. In 2 Round dataset training, we use ODE solver and Adam optimizer, with 2 hidden layer MLP, 100 hidden dimension, 800 batch size, 0.005 learning rate, and 1000 training steps. In 3 Round spin dataset training, we also use ODE solver and Adam optimizer, with 2 hidden layer MLP, 100 hidden dimension, 1000 batch size, 0.005 learning rate, and 2000 training steps. And in Dot-Circle dataset training, we use an ODE solver and Adam optimizer, with 2 hidden layer MLP, 100 hidden dimensions, 1600 batch size, 0.005 learning rate, and 10000 training steps.

## G.8 THIRD-ORDER HOMO

We optimize models by the sum of squared error(SSE). The source distribution and target distribution are all Gaussian distributions. For the first line, we use the original transport trajectory setting, followed by the VP ODE framework from (Liu et al., 2022b), which is $x_t = \alpha_t x_0 + \beta_t x_1$. We choose $\alpha_t = \exp(-\frac{1}{4}a(1-t)^2 - \frac{1}{2}b(1-t))$ and $\beta_t = \sqrt{1-\alpha_t^2}$, with hyperparameters $a = 19.9$ and $b = 0.1$. In 2 Round spin datasets and 3 Round spin datasets, we sample 400 points, both source distribution and target distribution. In the Dot-Circle dataset, we sample 600 points, both source distribution and target distribution, 300 points of source points from the circle, and another 300 from the center dot. In 2-round dataset training, we use an ODE solver and Adam optimizer, with 2 hidden layer MLP, 100 hidden dimensions, 800 batch size, 0.005 learning rate, and 1000 training steps. In 3-round spin dataset training, we also use an ODE solver and Adam optimizer, with 2 hidden layer MLP, 100 hidden dimensions, 1000 batch size, 0.005 learning rate, and 2000

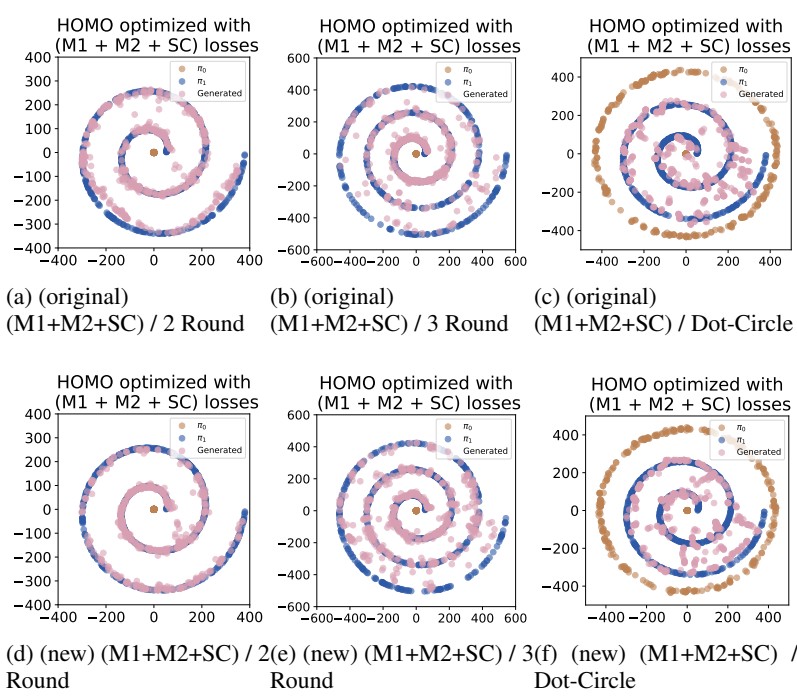

(a) (original)
(M1+M2+SC) / 2 Round

(b) (original)
(M1+M2+SC) / 3 Round

(c) (original)
(M1+M2+SC) / Dot-Circle

(d) (new) (M1+M2+SC) / 2 Round

(e) (new) (M1+M2+SC) / 3 Round

(f) (new) (M1+M2+SC) / Dot-Circle

Figure 19: (M1+M2+SC) The distributions generated by HOMO are only optimized by first-order loss and second order loss, and self-consistency loss. **Upper row(original trajectory setting):** Figure (a), in 2 Round spin dataset. Figure (b), in 3 Round spin dataset. Figure (c), in Dot-Circle dataset. **Lower row(new trajectory setting):** Figure (d), in 2 Round spin dataset. Figure (e), in 3 Round spin dataset. Figure (f), in Dot-Circle dataset. The source distribution, $\pi_0$ (**brown**), and the target distribution, $\pi_1$ (**indigo**), are shown, along with the generated distribution (**pink**).

training steps. In Dot-Circle dataset training, we use an ODE solver and Adam optimizer, with 2 hidden layer MLP, 100 hidden dimensions, 1600 batch size, 0.005 learning rate, and 10000 training steps.

## H COMPUTATIONAL COST AND OPTIMIZATION COST

Table 5: Computational Cost Analysis of Different Configurations

| Configuration | FLOPs (M) | Params (K) | Training Speed (it/s) |
|---|---|---|---|
| M1 | 8.400 | 10.702 | 477.03 |
| M2 | 68.480 | 43.608 | 146.34 |
| M3 | 8.400 | 10.702 | 357.45 |
| M1 + M2 | 16.960 | 21.604 | 248.15 |
| M2 + SC | 68.480 | 43.608 | 144.73 |
| (Shortcut Model) M1 + SC | 8.480 | 10.802 | 283.20 |
| M1 + M2 + SC | 68.480 | 43.608 | 136.46 |
| M1 + M2 + M3 + SC | 103.680 | 66.012 | 122.18 |

We profile computational efficiency on the Apple MacBook Air (M1 8GB) with an 8-core CPU. Through systematic analysis, we observe three critical tradeoffs: (1) The M2 configuration demonstrates an $8.15\times$ FLOPs increase over M1 while achieving $4.07\times$ parameter expansion, revealing the fundamental FLOPs-parameters scaling relationship. (2) The self-consistency (SC) term introduces minimal computational overhead, with the M2+SC configuration maintaining 144.73 it/s versus vanilla M2's 146.34 it/s (1.1% throughput reduction). (3) Architectural innovations yield

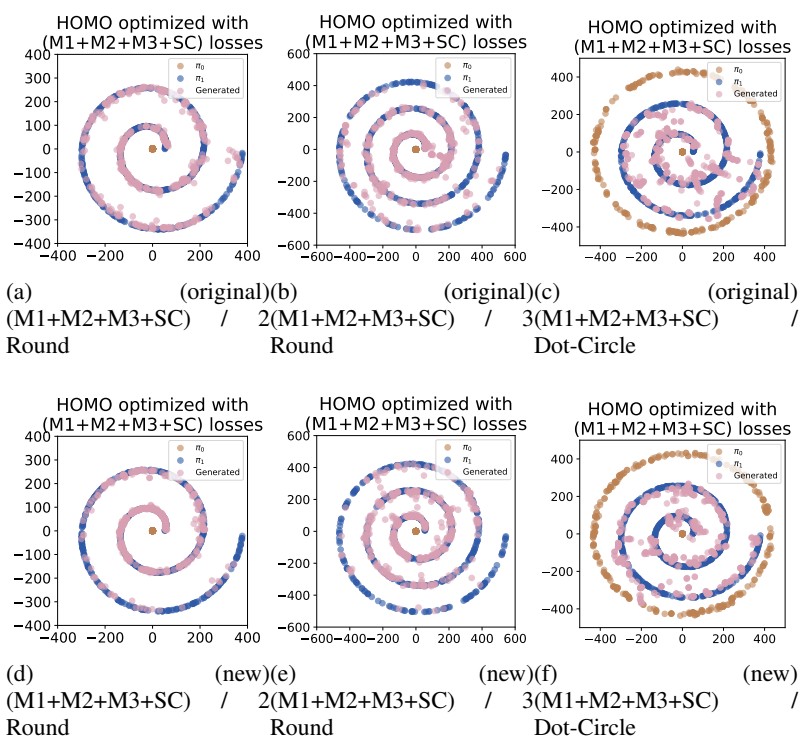

(a)                    (original)(b)                    (original)(c)                    (original)
(M1+M2+M3+SC)    /    2(M1+M2+M3+SC)    /    3(M1+M2+M3+SC)    /
Round                              Round                              Dot-Circle

(d)                    (new)(e)                    (new)(f)                    (new)
(M1+M2+M3+SC)    /    2(M1+M2+M3+SC)    /    3(M1+M2+M3+SC)    /
Round                              Round                              Dot-Circle

Figure 20: (M1+M2+M3+SC) The distributions generated by Third-Order HOMO, optimized by first-order loss and second-order loss, third-order loss and self-consistency loss. **Upper row(original trajectory setting):** Figure (a), in 2 Round spin dataset. Figure (b), in 3 Round spin dataset. Figure (c), in Dot-Circle dataset. **Lower row(new trajectory setting):** Figure (d), in 2 Round spin dataset. Figure (e), in 3 Round spin dataset. Figure (f), in Dot-Circle dataset. The source distribution, $\pi_0$ (**brown**), and the target distribution, $\pi_1$ (**indigo**), are shown, along with the generated distribution (**pink**).

substantial gains - the Shortcut Model (M1+SC) achieves 33.6% faster iterations than vanilla M1 (283.20 vs 477.03 it/s) with comparable parameter counts. Table 5 quantifies these effects through comprehensive benchmarking:

Notably, our architecture maintains practical viability even for high-order extensions - the third-order HOMO configuration (M1+M2+M3+SC) sustains 122.18 it/s despite requiring $12.34\times$ more FLOPs than the base M1 model. This demonstrates our method's ability to balance computational complexity with real-time performance requirements.

## LLM USAGE DISCLOSURE

LLMs were used only to polish language, such as grammar and wording. These models did not contribute to idea creation or writing, and the authors take full responsibility for this paper's content.

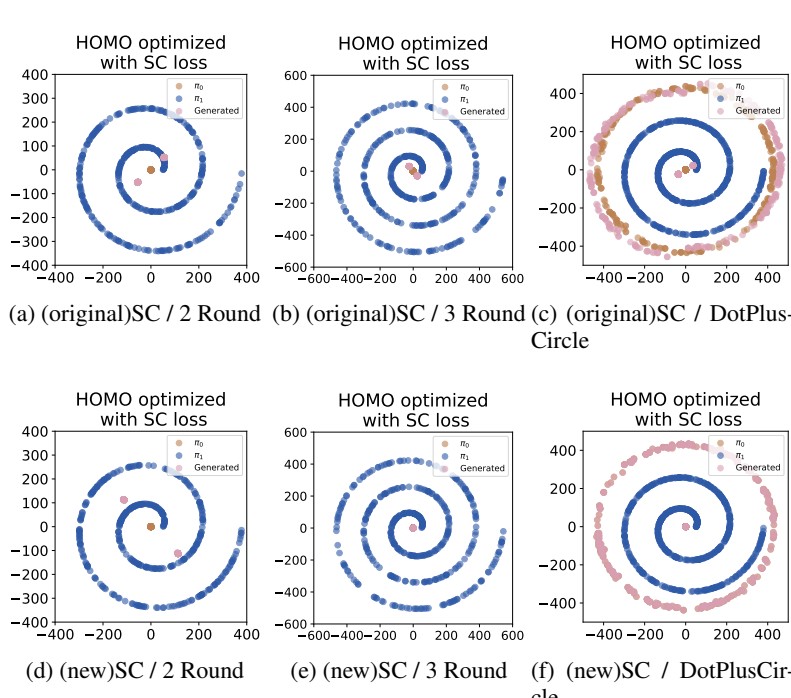

(a) (original)SC / 2 Round  (b) (original)SC / 3 Round  (c) (original)SC / DotPlus-Circle

(d) (new)SC / 2 Round    (e) (new)SC / 3 Round    (f) (new)SC / DotPlusCircle

Figure 21: (SC ) The distributions generated by HOMO are only optimized by self-consistency loss. **Upper row(original trajectory setting):** Figure (a), in 2 Round spin dataset. Figure (b), in 3 Round spin dataset. Figure (c), in Dot-Circle dataset. **Lower row(new trajectory setting):** Figure (d), in 2 Round spin dataset. Figure (e), in 3 Round spin dataset. Figure (f), in Dot-Circle dataset. The source distribution, $\pi_0$ (**brown**), and the target distribution, $\pi_1$ (**indigo**), are shown, along with the generated distribution (**pink**).

