# OpenReview forum: "High-Order Matching for One-Step Shortcut Diffusion Models"
_ICLR.cc/2026/Conference — ICLR 2026 Conference Withdrawn Submission_

### Official Review · Reviewer_k7cZ · 2025-10-31

**Soundness:** 2
**Presentation:** 2
**Contribution:** 3
**Rating:** 2
**Confidence:** 3

**Summary:**

The paper proposes an extension of shortcut diffusion models: HOMO which allows for the inclusion of higher-order terms in the objective to improve the ability of learned models to capture flow trajectories.
Furthermore, the paper provides theoretical justification for the importance of incorporating higher-order terms and evaluates HOMO on 2D toy distributions.

**Strengths:**

* The main idea behind the paper is well-founded and motivated, as it makes a lot of sense to expect that further guidance from higher order terms should help performance.
* The theoretical justification for the use and importance of higher order terms in HOMO is, as far as I can see, good.
* The paper does include a lot of empirical results, from providing many ablations with different combinations of M1, M2, SC.

**Weaknesses:**

* The main issue with the paper is that all reported results are on 2D toy experiments. For instance, there are no large scale evaluations on problems of actual interest such as CIFAR or ImageNet, as well as baselines with other relevant methods on one-step/few-step generative models, like [1, 2] etc.

* The main point that the paper needs to make is that the computational trade-off of additional computation from handling higher-order term, pays off in terms of performance. From Appendix H, there is a decent analysis of this. However, this is critically limited as the reported numbers are only for CPU performance, whereas practically, we need information about GPU performance.

* The language used in the paper is way too overstated, such as "a game-changing framework that leverages high-order supervision to revolutionize distribution transportation" (line 19) and "setting a new standard for one-step generative models" (line 26) etc., especially, since the empirical evaluation of HOMO is so limited that it does not actually justify such statements

* I find the inclusion of Appendix B on LLMs to be very strange, since as far as I can see, this section bears no relevance to the actual contents of the paper. This needs to be removed.

[1] Consistency Models (2023)
[2] Inductive Moment Matching (2025)

**Questions:**

* How sensitive is performance from the choice of discretisation of d?
* Did you explore any weighting schemes for the new collection of different loss terms?
* Do you believe that the trend reported in the current empirical results will be able to translate to large scale datasets of practical interest?
* Why did you include Appendix B?

---

> ### Author Response · Authors · 2025-11-26
>
> Thank you very much for your valuable review.  We will update in our next version.

---

### Official Review · Reviewer_q2v7 · 2025-11-01

**Soundness:** 3
**Presentation:** 2
**Contribution:** 1
**Rating:** 2
**Confidence:** 4

**Summary:**

This paper takes the paper of [Frans et al. 2025] cited in the abstract, and updates it to add learnable second order taylor expansions rather than just the velocity fields of the particles. It has also some convergence bounds, which adapt proofs from a previous work by [Fukumizu et al. 2024] and some toy case experiments confirming that Taylor series of second order approximate better than first order.

**Strengths:**

The main strength is that the paper has a good idea, namely to use second order taylor approximation in the setting of [Frans et al. 2025], however I think that this paper is not in final form.

**Weaknesses:**

The main weakness is the experiments. If the claim is that second or higher order is better than the original, then it should be tested on the same datasets and prove stronger performance there.

Testing on 2D distributions does not convince very much, because we all know that higher dimensional geometry has sometimes counterintuitive properties that are not easily captured by just testing on distributions (as complex as they may be called) in R^2.

The proofs of the main theorems (appendices. C, D) are not a big step away from the work [Fukumitzu et al. 2024] but this is fair enough, if a big step is not needed then why do it.

Also, the paper has a series of typos and weird formulations, so can you re-read it a bit, and maybe pass it through a spellchecker?
Some examples:
- line 106: "knowledge of only $x_t$ renders it a random variable" what do you mean? it's not the knowledge of $x_t$ that matters, but the fact that $x_0$ is random, no?
- line 114 "the denoising ODEs" .. are there more than one ODE?
- line 132 "gradient" and "second-gradient" sounds weird.. you mean time derivative and second time derivative right?
- line 133 "reprectively"
- in all definitions you define $\Delta t = 1/128" with no explanation, and then take d to belong to a tuple, sometimes including 0 sometimes not.. based on what criterion I don't understand
- in definition 3.2 the smallest allowed value of d is 1/128, but then the condition d<1/128 appears in line 144.. I don't follow much, can you revise/explain?
- in definition 4.1 the only allowed value of d<1/128 is d=0 right? and you take $\Delta t =1/128$ and you then stipulate when defining $x_{t+d}$ that when d<1/28 (i.e. when d=0 which was the only choice smaller than 1/128) you replace that by 1/128.. this is overwhelmingly strange.. can you explain?
- line 147: "first order term" of what? you never talked about terms before..
- Def. 4.2: "Let $u_{1,\theta_1}$ be the networks" -- it's only one network though no?
- Remark 4.4: "we denote first-order matching as M1, which implies that HOMO is optimized solely by the first-order loss".. this has to be reworded, it's not the fact that you denote something that implies that HOMO is optimized in a way or another.. also, it's not optimized by some loss, is trained to optimize some loss I guess?

I stopped annotating typos after arriving at section 5, but there may be more, please have a check..

**Questions:**

Can you compare your method to others on more realistic datasets/tasks?

What are the drawbacks in terms of compute time or in terms of scaling, for the higher order methods you propose? Can you comment/compare on these, in case you decide to do more realistic experiments?

---

> ### Author Response · Authors · 2025-11-26
>
> Thank you for your helpful feedback. We will update the paper in the next revision.

---

### Official Review · Reviewer_r58d · 2025-11-02

**Soundness:** 1
**Presentation:** 1
**Contribution:** 2
**Rating:** 2
**Confidence:** 3

**Summary:**

The paper proposes HOMO, a higher-order Shortcut diffusion model that augments first-order (velocity) supervision with acceleration (and optionally jerk) along $x_t = \alpha_t x_0 + \beta_t x_1$. Each order is predicted by its own network and trained with losses for first/second-order matching plus a self-consistency constraint from composing small steps. They provide training/sampling schemes and show experiments on synthetic and curve datasets.

**Strengths:**

The core idea of explicitly modeling higher-order terms along the transport path with separate networks is simple and potentially broadly applicable. The paper gives concrete training and sampling procedures that are easy to implement, and on standard 2D benchmarks it shows consistent improvements over a first-order shortcut baseline.

**Weaknesses:**

The writing quality materially hurts readability. For example, the text says "we define Shortcut model compute next field" which is ungrammatical and obscures meaning. On the theory side, the approximation bounds are not informative for learning: even with large models, the bound in 5.1 retains an additive term $\mathbb{E} \left[\|\dot x_{\text{true}}-\ddot{x}_{\text{true}}\|^2\right]$ that does not vanish. The results do not show that the learned velocity and acceleration converge to the truth nor that the minimizer of the proposed loss recovers a correct generative model of the data. The loss design is also unconvincing. If the Shortcut first order objective is optimized to match the path velocity, no second order correction should be needed, and the paper does not explain why adding a second order term helps. In addition there is likely an error in the objective as written since the M1 loss appears to evaluate $u_1(x_t,t,2d)$ rather than the instantaneous argument. Finally, the empirical scope is narrow since there are no image or other high dimensional experiments, so it is unclear whether any gains extend beyond two dimensional toys.

**Questions:**

1) The approximation bounds include a non-vanishing term $\mathbb{E} \left[|\dot x_{\text{true}}-\ddot{x}_{\text{true}}|^2\right]$. Could the authors clarify how this bound provides any guarantee that the learned model converges to the true generative process or that the proposed losses identify the correct flow?

2) Why is the first-order M1 loss evaluated with $u_1(x_t,t,2d)$ rather than at the instantaneous argument $u_1(x_t,t,0)$? If this is intentional, what theoretical justification ensures that using a finite step size does not bias the learned dynamics?

3) All reported experiments are on 2D toy datasets. Have the authors tested the approach on any image or high-dimensional generative tasks, and if not, what challenges prevent such evaluation?

---

> ### Author Response · Authors · 2025-11-26
>
> Thank you very much for your helpful comments. We will incorporate them in the next revision.

---

### Note · Authors · 2025-11-26

I have read and agree with the venue's withdrawal policy on behalf of myself and my co-authors.